# SpyMask enables combinatorial assembly of bispecific binders

Claudia L. Driscoll [1,2], Anthony H. Keeble[1,2] & Mark R. Howarth [1,2] ✉

Bispecific antibodies are a successful and expanding therapeutic class. Standard approaches to generate bispecifics are complicated by the need for disulfide reduction/oxidation or specialized formats. Here we present SpyMask, a modular approach to bispecifics using SpyTag/SpyCatcher spontaneous amidation. Two SpyTag-fused antigen-binding modules can be precisely conjugated onto DoubleCatcher, a tandem SpyCatcher where the second SpyCatcher is protease-activatable. We engineer a panel of structurally-distinct DoubleCatchers, from which binders project in different directions. We establish a generalized methodology for one-pot assembly and purification of bispecifics in 96-well plates. A panel of binders recognizing different HER2 epitopes were coupled to DoubleCatcher, revealing unexpected combinations with anti-proliferative or pro-proliferative activity on HER2-addicted cancer cells. Bispecific activity depended sensitively on both binder orientation and DoubleCatcher scaffold geometry. These findings support the need for straightforward assembly in different formats. SpyMask provides a scalable tool to discover synergy in bispecific activity, through modulating receptor organization and geometry.

Bispecific antibodies are biological matchmakers, bringing together two different target molecules or two different parts of the same target[1,2]. Bispecifics have had clinical success in cancer therapy by directing cytotoxic T cells to cancer-specific antigens[1,2]. There are also approved bispecific therapeutics that bridge proteins on the surface of the same cell[3,4]. Cell signaling often naturally depends on co-localization of particular ligands at the cell-surface[5]. Synthetic control of such signaling has been successful using bifunctional small molecules[6], DNA nanostructures[7], and non-antibody protein-based binders[8]. The response of the cell is often exquisitely sensitive to the timing and spatial orientation of this bridging between surface targets[9]. Therefore, despite the numerous successes from bispecifics, there remains a huge range of unexplored combinations of bispecifics that could illuminate the principles of cell signaling and provide valuable therapeutic leads. It is an exciting challenge in protein engineering to facilitate the screening of bispecific combinations.

The most common routes to generate bispecifics depend upon the rearrangement of IgG architecture, particularly through knob-into-hole generation of heterodimeric Fc domains[10], or disulfide bond shuffling for Fab-arm exchange[11–13], leading to many important clinical successes[2]. Alternative bispecific pairing has been achieved by conjugating the two binders using sortase[14], transglutaminase[15], click chemistry[16] or split inteins[17]. However, these routes require each binder to be fused to the respective conjugation domain, such that each binder must be re-cloned, re-expressed, and re-purified in two formats to explore all possible bispecific combinations. Sophisticated combinatorial DNA libraries have been employed to impart automation into the bispecific library production pipeline[18], although these methods are only suited to single-chain bispecific molecules. Furthermore, while it is widely appreciated that subtle differences in bispecific format parameters can exhibit profound and unpredictable changes in activity[19], few assembly platforms to date have been designed to optimize more parameters than binder identity and the relative orientation of binders within the bispecific molecule[18,20,21]. As the field moves towards exploring combinations of thousands of binders across the proteome (with potentially millions of bispecifics arising), there is

[1]Department of Biochemistry, University of Oxford, South Parks Road, Oxford OX1 3QU, UK. [2]Department of Pharmacology, University of Cambridge, Tennis Court Road, Cambridge CB2 1PD, UK. ✉e-mail: mh2186@cam.ac.uk

an opportunity for new bispecific antibody assembly platforms to allow screening of multiple format parameters simultaneously, without dramatically increasing the manufacturing burden[22–24].

Our group previously developed an efficient way to ligate genetically encoded building blocks, through spontaneous isopeptide bond formation between a peptide Tag and a complementary Catcher protein[25,26]. SpyTag/SpyCatcher has been widely used for the modular assembly of binders, enzymes and vaccines[25,27]. SpyTag003/SpyCatcher003 is a pair that we subsequently developed for reaction at a rate close to the diffusion limit, which maintains compatibility with all previous Spy versions[28]. SpyTag/SpyCatcher has been previously employed for bispecific assembly but faces the above limitation of having to clone each binder in two different formats, with one version fused to SpyTag and the other version fused to SpyCatcher[26,29,30].

In this work we establish a protease-activatable SpyCatcher003, to enable the generation of bispecific combinations where each binder only needs to be cloned into one common format – the SpyMask strategy. We show that binders can be assembled onto a panel of scaffolds, in order for the binders to project in different directions and change the orientation of the target receptor. We evaluate the SpyMask system against human epidermal growth factor receptor 2 (HER2), which has a central role in breast, gastric, and ovarian cancers through proliferative and anti-apoptotic signaling[31]. Employing Spy-Mask on a panel of 8 binders (affibody, nanobody or antibodies) allowed the simple generation of 64 binder pairs against HER2 (8 with bivalent and 56 with bispecific binding activity). In order to demonstrate the versatility of our modular bispecific assembly approach, DoubleCatcher scaffolds that vary in dimensional architecture and rigidity were designed by modifying the linker separating Catcher domains, or by introducing an intramolecular disulfide bond. Again, SpyMask was applied to dimerize a subset of anti-HER2 binders using the panel of 7 DoubleCatcher variants, to generate 63 bivalent binders spanning a diverse format space. These pairs of binders show major differences in their effect on cell survival and division, illustrating the potential of the SpyMask approach to accelerate combinatorial discovery and fine-tune cellular responses to designed receptor modulators.

## Results

### Masked SpyCatcher003 is a protease-activatable variant of SpyCatcher003

Our approach uses a tandem SpyCatcher003 for specific irreversible heterodimerization of any two antigen-binding units fused to SpyTag or its variants (i.e. SpyTag002 or SpyTag003) (Fig. 1A). The N-terminal SpyCatcher003 is accessible for reaction. The C-terminal Spy-Catcher003 is masked. The non-reactive SpyTag003 D117A (Spy-Tag003DA) mutant[28] is fused to the C-terminus of the second SpyCatcher003 via a flexible linker containing a tobacco etch virus (TEV) protease cleavage site (ENLYFQ/G) (Fig. 1A). Before cleavage, SpyTag003DA should bind tightly to SpyCatcher003, with $K_d = 21$ nM for the non-linked pair[28], thus masking the reactive Lys of Spy-Catcher003. The intramolecular SpyCatcher003/SpyTag003DA interaction should be entropically favored over a competing intermolecular interaction with SpyTag003 in solution. Following cleavage at the TEV site, the SpyTag003DA peptide will be free to dissociate, unmasking the reactive Lys of SpyCatcher003 to enable reaction with a second supplied SpyTag-linked binder. We performed all cleavages with superTEV protease, because of its efficient activity without the need for reductant[32].

We piloted the SpyMask strategy on a single SpyCatcher003 moiety. Masked SpyCatcher003 was cleaved with superTEV protease and incubated with a model fusion protein of SpyTag003 linked to maltose-binding protein (MBP) (Fig. 1B). This reaction proceeded efficiently to > 99% completion, with a half-life of reaction ($t_{1/2}$) of $2.7 \pm 0.7$ min (mean $\pm 1$ s.d., $n = 3$) (Fig. 1B–D). To reduce the extent to

which SpyTag003DA dissociation limits the rate of reaction between TEV-cleaved Masked SpyCatcher003 with SpyTag003-MBP, the reaction temperature was elevated to 37 °C and the molar excess of SpyTag003-MBP relative to Masked SpyCatcher003 was increased to 5-fold. The muted reactivity of Masked SpyCatcher003 before cleavage was probed by incubating equimolar amounts of the construct with SpyTag003-MBP and monitoring the disappearance of Masked SpyCatcher003 over time by SDS-PAGE and densitometry (Fig. 1D, E). Under these conditions, we observed 3.7% reaction of Masked Spy-Catcher003 with SpyTag003-MBP after 60 min, supporting effective silencing of SpyCatcher003 reactivity.

### Establishing DoubleCatcher as a tool for controlled combination of distinct proteins

To design a platform that facilitates the conjugation of two Tagged proteins, Masked SpyCatcher003 was genetically fused to the C-terminus of a native SpyCatcher003 protein via a flexible Gly/Ser linker, to give DoubleCatcher (Fig. 2A, with sequence shown in Supplementary Fig. 1). The reactivity of each SpyCatcher003 moiety within the DoubleCatcher construct was determined by monitoring the reaction of DoubleCatcher with SpyTag003-linked model proteins before and after cleavage by MBP-superTEV protease. First, DoubleCatcher was incubated with 2-fold excess of SpyTag003 linked to the green fluorescent protein mClover3 in the absence of TEV protease (Fig. 2A, B). 91% reactivity of the N-terminal SpyCatcher003 was observed after 1 h, with a $t_{1/2}$ of $0.58 \pm 0.05$ min (mean $\pm 1$ s.d., $n = 3$) (Fig. 2C).

Next we incubated DoubleCatcher with two-fold excess SpyTag003-MBP for 2 h to saturate the N-terminal SpyCatcher003 moiety. Unreacted SpyTag003-MBP was removed from the reaction mixture by overnight incubation with bead-immobilized Spy-Catcher003. Addition of superTEV protease to the crude mixture (4 h, 34 °C) induced cleavage of the SpyTag003DA mask from the C-terminal SpyCatcher003 (Fig. 2D). 2 μM of the cleaved construct was then reacted with a 5-fold molar excess of SpyTagged Small Ubiquitin-like Modifier (SUMO-SpyTag003) and the reaction was monitored over time, by SDS-PAGE and densitometry (Fig. 2E, F). Reaction to fill the second site of DoubleCatcher was rapid with $t_{1/2}$ of $3.6 \pm 0.7$ min (mean $\pm 1$ s.d., $n = 3$) and proceeded to >98% completion, showing that the cleaved SpyTag003DA can be efficiently replaced by a reactive Tag.

### Establishment of scalable heterodimerization of proteins using SpyMask

A generally applicable bispecific assembly platform for high-throughput screening requires a simple and scalable methodology. In particular, procedures involving multiple rounds of column chromatography will provide a bottleneck to the combinatorial complexity of bispecifics that can be interrogated. Therefore, we next sought to optimize a one-pot reaction in which any two Tagged binders can be conjugated to DoubleCatcher (Fig. 3A), and the resulting bispecific molecule can be purified (Fig. 3B). For optimization, model ligands were assembled onto DoubleCatcher: these were a SpyTag construct linked to three tandemly-repeated affibodies against HER2 (SpyTag-AffiHER2₃ which we abbreviate to ligand A) and SpyTag003 linked to a high stability MBP tandem fusion, $(MBP_X)_2$-SpyTag003 which we abbreviate to ligand B[33]. These ligands lack free Cys (avoiding impurity from inter-ligand disulfides) and have a difference in $M_w$ that permits unambiguous detection of reactant and product bands by SDS-PAGE (Fig. 3A). The intact masses of these purified reactants in phosphate-buffered saline (PBS) pH 7.4 were validated by electrospray-ionization mass spectrometry (Supplementary Fig. 2).

In step 1 of our one-pot assembly route, DoubleCatcher reacts with a small molar excess (1.25-fold) of SpyTag-AffiHER2₃, to ensure negligible unreacted DoubleCatcher in solution (Fig. 3A). SDS-PAGE revealed 97% of the reaction contained the desired singly-occupied

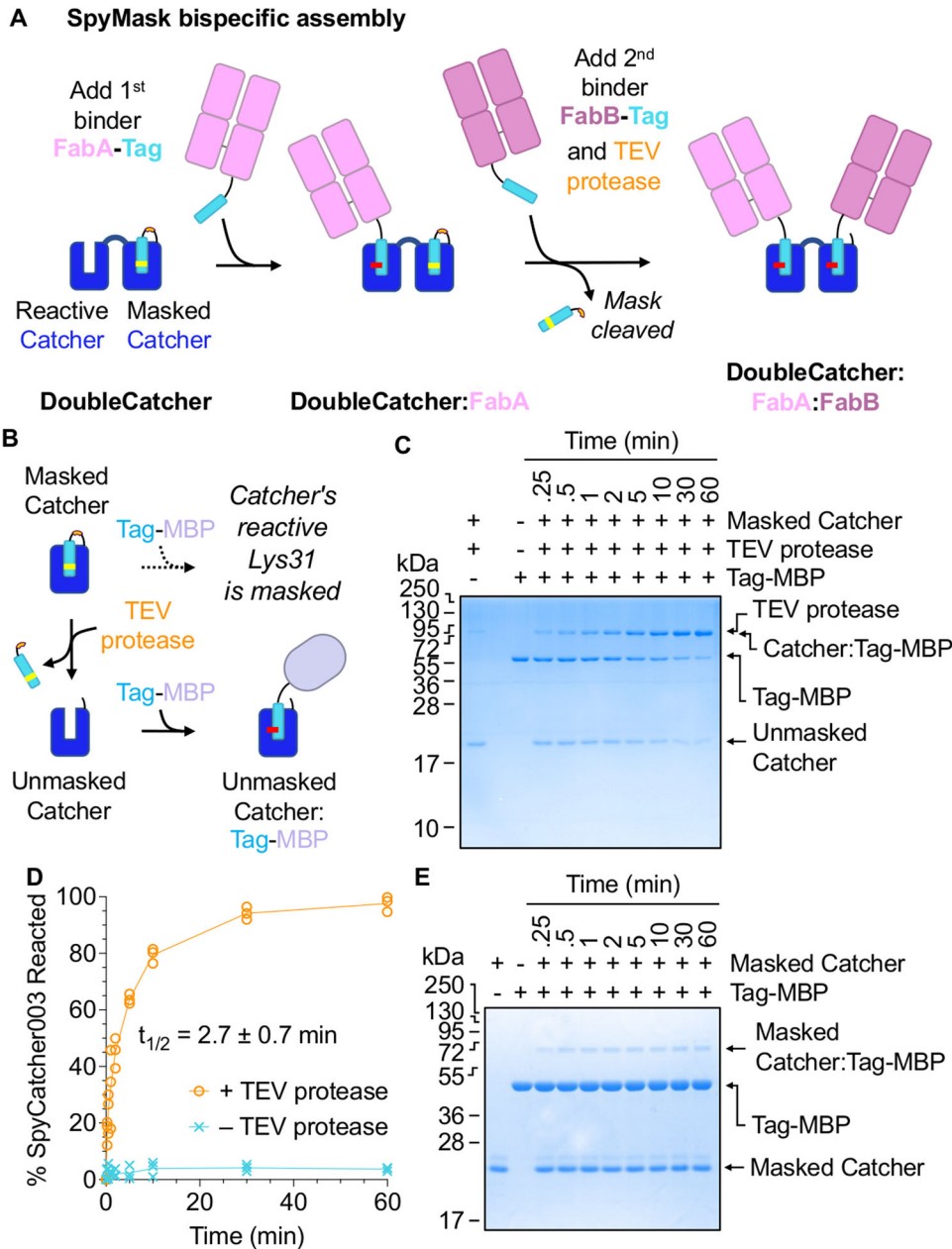

**Fig. 1 | Masked SpyCatcher003 has protease-activatable reactivity. A** SpyMask approach. DoubleCatcher consists of an accessible N-terminal SpyCatcher003 (Reactive Catcher) and a C-terminal SpyCatcher003 masked by the unreactive SpyTag003DA peptide (Masked Catcher). The first SpyTag-linked binder reacts with the N-terminal Catcher but is blocked from reacting at the second Catcher by the tethered unreactive SpyTag003DA peptide. Addition of TEV protease induces release of SpyTag003DA, so that the second Catcher can be loaded with a different SpyTag-linked binder. SpyTag003 in cyan with D117A indicated in yellow, TEV protease site in orange, Fabs in pink, and isopeptide bond in red. A colon indicates covalently linked products. **B** Cartoon following protease-controlled unmasking of SpyCatcher003. Incubation of Masked Catcher with TEV protease enables cleavage and dissociation of SpyTag003DA from SpyCatcher003, allowing SpyTag003-linked proteins to react with SpyCatcher003 (Unmasked Catcher). **C** Efficient coupling after TEV protease unmasking. After incubation of Masked Catcher with TEV protease, the Unmasked Catcher ($5\,\mu$M) is incubated with equimolar SpyTag003-MBP (Tag-MBP) for the indicated time, before SDS-PAGE with Coomassie staining ($n = 3$ independent time-course reactions; 1 gel per reaction). **D** Quantification of SpyCatcher003 reactivity with or without masking (individual replicates with means connected, $n = 3$ independent reactions), indicating the mean half-time of reaction, $t_{1/2}$ (shown ± 1 s.d., $n = 3$ independent reactions). **E** Minimal reactivity of Masked Catcher prior to TEV protease cleavage. Masked Catcher was incubated with Tag-MBP (both partners at $5\,\mu$M) for the indicated time at 37 °C, before SDS-PAGE/Coomassie ($n = 1$ gel per independent time-course reaction). Source data are provided as a Source Data file.

DoubleCatcher:SpyTag-AffiHER2₃ species (Fig. 3C, lane DC + A). As observed for Masked SpyCatcher003 reactivity tests (Fig. 1D), 3% contained DoubleCatcher conjugated to two SpyTag-AffiHER2₃ molecules (Fig. 3C). To confirm where the homodimeric DoubleCatcher:(SpyTag-AffiHER2₃)2 would run, we added superTEV protease (1/12ᵗʰ molar equivalent) to the crude mixture; the homodimer in the reaction mixture increased to 30% ('DC + A + TEV' lane in

Fig. 3C). The identity of this by-product was corroborated by the increase in the density of the band corresponding to AffiHER2₃ homodimer in a control reaction containing twice the concentration of SpyTag003-AffiHER2₃ ('DC + A + TEV + A' lane in Fig. 3C).

For subsequent unmasking and saturation of the C-terminal SpyCatcher003 moiety, superTEV protease (1/12ᵗʰ molar equivalent) and (MBPₓ)₂-SpyTag003 (5-fold molar excess, 'B') were added

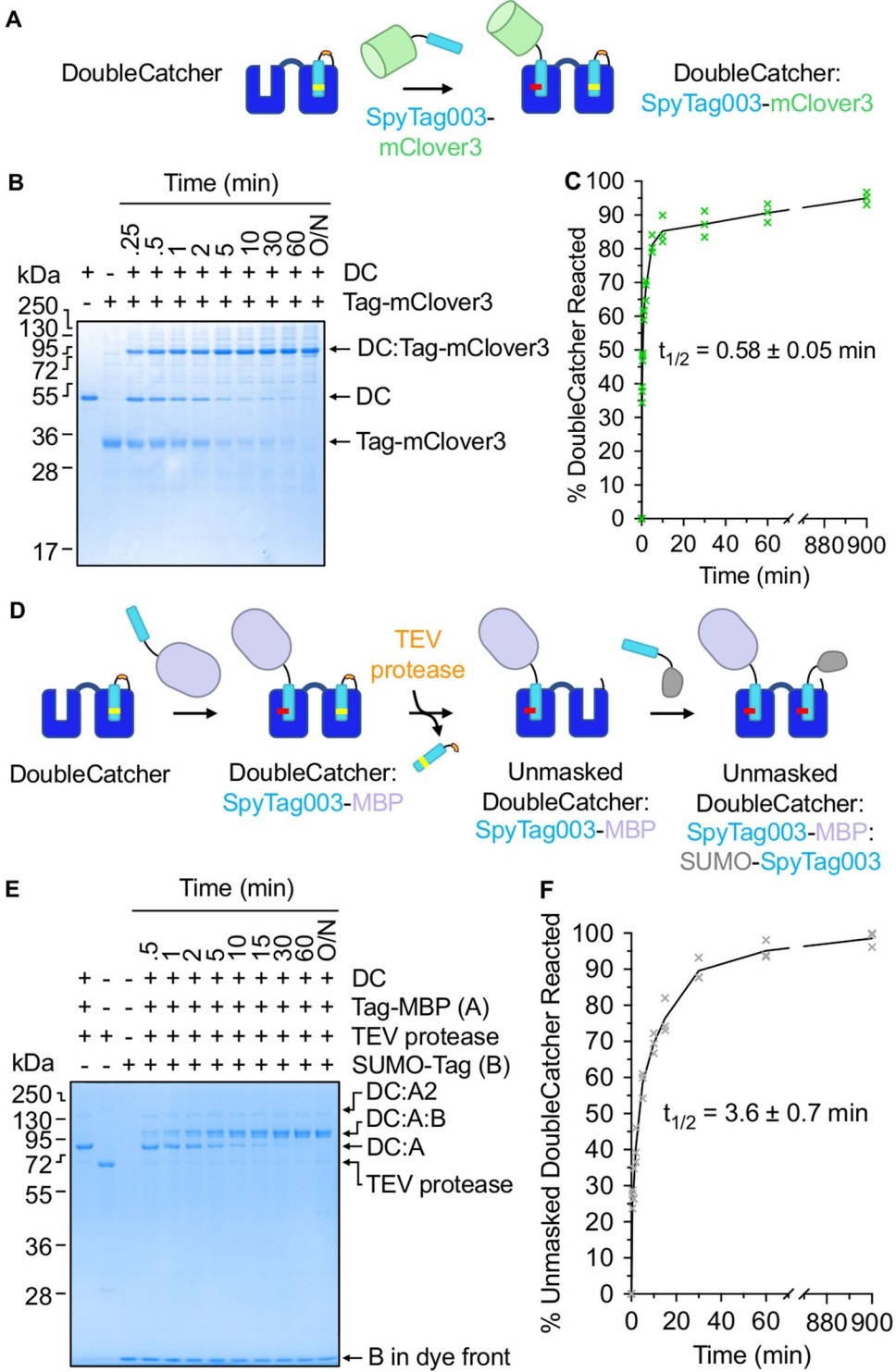

**Fig. 2 | Dual conjugation of Tagged proteins to DoubleCatcher is fast and site-specific. A** Schematic of covalent conjugation of SpyTag003-mClover3 (Tag-mClover3) to the reactive N-terminal Catcher of DoubleCatcher. **B** Time-course of reaction between DoubleCatcher (DC; 2.5 μM) and SpyTag003-mClover3 (Tag-mClover3; 5 μM), analyzed by SDS-PAGE/Coomassie (*n* = 3 independent time-course reactions; 1 gel per reaction). **C** Depletion of unreacted DoubleCatcher over the course of reaction was quantified by densitometry, following (**B**) ($t_{1/2}$ mean ± 1 s.d., *n* = 3 independent reactions). **D** Schematic depicting unmasking and conjugation of the C-terminal Catcher of DoubleCatcher. **E** Reaction at the unmasked

Catcher of DoubleCatcher. Following saturation of the N-terminal Catcher of DoubleCatcher (DC) with SpyTag003-MBP (**A**), the sample was incubated with TEV protease for 2 h at 34 °C. The resulting unmasked DoubleCatcher:SpyTag003-MBP (2 μM) was then reacted with SUMO-SpyTag003 (B; 10 μM) over the time indicated, before SDS-PAGE/Coomassie (*n* = 3 independent time-course reactions; 1 gel per reaction). **F** Quantification of reaction at the unmasked Catcher of DoubleCatcher, based on **E**. All triplicate data points are shown, with the line connecting the mean. ($t_{1/2}$ mean ± 1 s.d., *n* = 3 independent reactions). Source data are provided as a Source Data file.

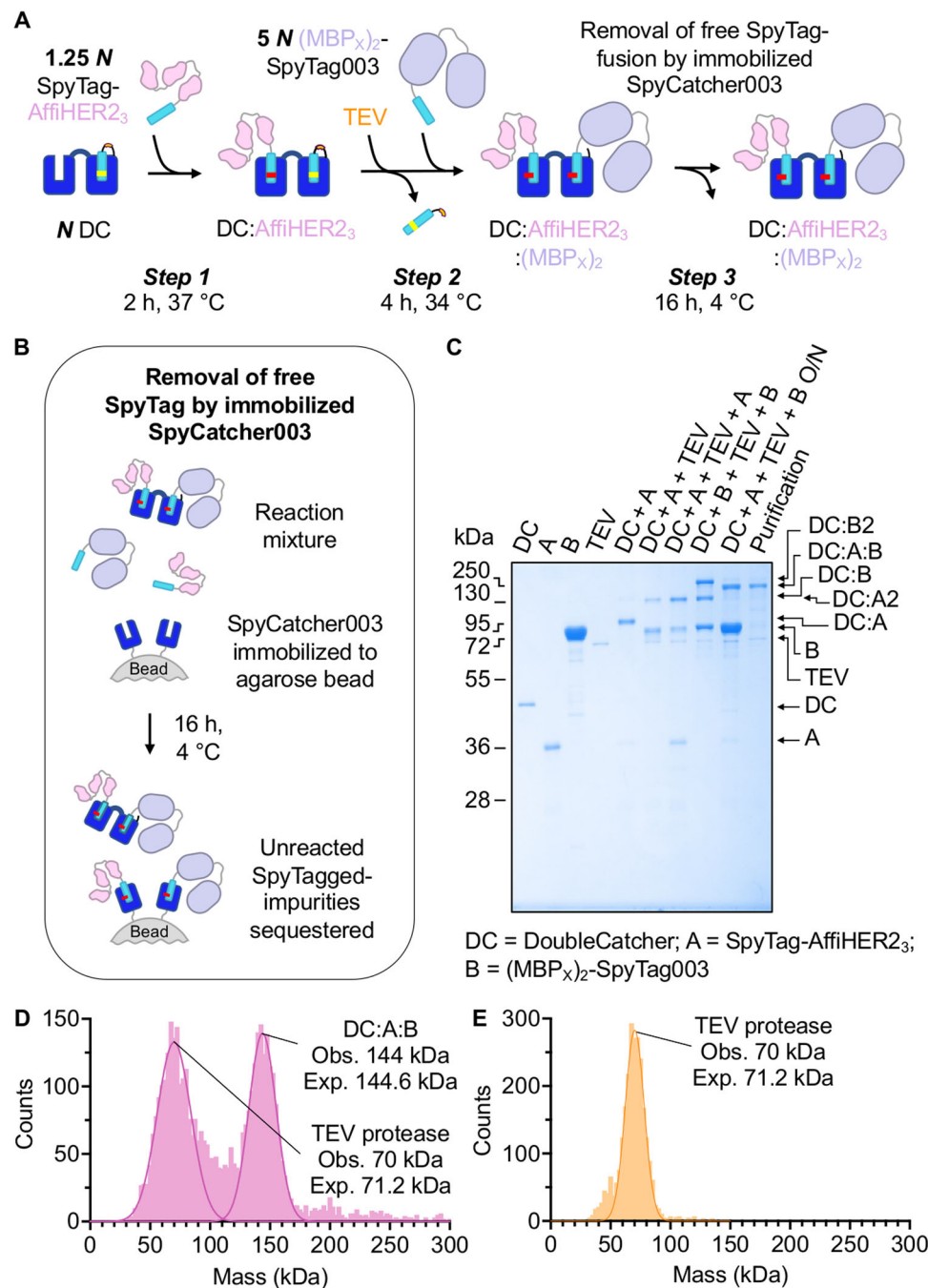

**Fig. 3 | One-pot conjugation of DoubleCatcher with Tagged model proteins to yield a bispecific with high purity.** (**A**) Schematic depicting assembly of SpyTag-AffiHER2₃ (pink) and (MBPₓ)₂-SpyTag003 (purple) onto DoubleCatcher (DC, blue). *N* represents the concentration of DoubleCatcher and then the relative molarity of each Tag-binder is marked. Immobilized SpyCatcher003 removes free Tagged proteins, leaving the bispecific in solution. **B** Schematic of purification of excess Tag-fusion by immobilized SpyCatcher003 S49C in Step 3. **C** One-pot bispecific assembly as in **A**, monitored by SDS-PAGE/Coomassie. DC = DoubleCatcher; A = SpyTag-AffiHER2₃; B = (MBPₓ)₂-SpyTag003 (*n* = 1 assembly reaction). **D** Mass photometry of SpyMask reaction. Peaks are shown for the product of coupling, as in **B**, with observed and expected M_w indicated, confirming the identity of the target bispecific. The other species in the reaction was TEV protease. **E** Mass photometry of TEV protease alone, annotated as in **D**. Source data are provided as a Source Data file.

simultaneously to the singly saturated DoubleCatcher (*Step 2*, Fig. 3A). Thus, immediately upon cleavage of SpyTag003DA, the concentration of (MBPₓ)₂-SpyTag003 in solution should be 20 times that of unreacted SpyTag-AffiHER2₃ and so homodimer formation is minimized. Indeed, AffiHER2₃ homodimer accounted for only 3% of the total species in the reaction mixture ('DC + A + TEV + B' in Fig. 3C). Trace (3%) formation of DoubleCatcher conjugated to two (MBPₓ)₂-Spy-Tag003 molecules ('MBP homodimer') was observed by SDS-PAGE, validated by the formation of a band of equivalent mobility in a

reaction with DoubleCatcher, superTEV protease and (MBPₓ)₂-Spy-Tag003 alone ('DC + B + TEV + B' lane in Fig. 3C).

For depletion of excess SpyTag- or SpyTag003-linked binder from the desired product, the crude reaction mixture was incubated with bead-immobilized SpyCatcher003, by adding beads in a 1:1 suspension with PBS directly to the reaction mixture (*Step 3*, Fig. 3A and Fig. 3B). The purity of DoubleCatcher:SpyTag-AffiHER2₃:(MBPₓ)₂-SpyTag003 following purification was assessed by SDS-PAGE densitometry as 95% (excluding superTEV protease) (Purification lane in Fig. 3C). Based on

SDS-PAGE, superTEV protease (70 kDa, Fig. 3D, E) was the major contaminant. Since TEV protease has exceptional specificity for its target sequence, which is not found in the human proteome, and has been expressed before without effect on cells[34], we did not consider it necessary to remove. However, it should also be possible to add amylose-agarose to the well, for depletion of the MBP-linked superTEV protease, if considered helpful.

We also validated the bispecific assembly by mass photometry, allowing single molecule-based assessment of each species in solution[35]. The peak corresponding to DoubleCatcher:SpyTag-AffiHER2$_3$:(MBP$_X$)$_2$-SpyTag003 was observed with a mass of 144 kDa, matching well to the 144.6 kDa predicted mass (Fig. 3D). The second peak of 70 kDa, fitted well to MBP-superTEV protease (Fig. 3D), as validated by running MBP-superTEV protease on its own (Fig. 3E).

## Anti-HER2 bispecifics assembled using SpyMask induce various anti-proliferative or proliferative effects in HER2-addicted cancer cells

The work presented above demonstrates the efficient assembly of bispecific ligands through DoubleCatcher. We then aimed to complement the SpyMask platform with a screening approach of equal throughput on binders against the important cell-surface target of HER2. The relative activity of each signaling pathway and downstream physiological outcome has been attributed to different ligands stabilizing receptor dimers in distinct conformations, inducing distinct phosphorylation of the 17 sites in HER2's intracellular domain[36]. Although the functional selectivity of native Receptor Tyrosine Kinase ligands has been extensively studied, this bias is yet to be exploited for personalized cancer therapies. Thus, we sought to assemble a large matrix of anti-HER2 bispecifics, which differ in format and screen for their ability to alter HER2 activity[37,38]. Binders specific to each subdomain of the HER2 extracellular region were identified (Fig. 4A, B), including Fabs based on the clinically used Pertuzumab ('Pert', known commercially as Perjeta) and Trastuzumab ('Tras', known commercially as Herceptin). Here we used a Trastuzumab variant with a point mutation to enhance HER2 binding stability[39]. Each binder was expressed as a SpyTag003-fusion in Expi293F cells, except the nanobody (nanoHER2)[40] and affibody (AffiHER2)[41] which were expressed in *Escherichia coli*. We confirmed the identity of each binder using electrospray-ionization mass spectrometry (Supplementary Fig. 3). Fusions to SpyTag are conventionally expressed with a spacer to favor high reactivity with SpyCatcher and optimal activity of the fused domain[42]. We expressed a variant of Tras with no linker to SpyTag003, to test the importance of linker flexibility on activity, termed Tras NoLink. The NoLink variant was confirmed to react to 97% in 60 min in PBS pH 7.4 at 37 °C with DoubleCatcher, an efficiency comparable to the linker-containing parental construct (99% in 60 min under equivalent reaction conditions; Supplementary Fig. 4), confirming that reactivity is not impaired by removing the linker between the Fab and the reactive peptide.

The interaction strength of each binder on recombinantly-expressed HER2 extracellular domain (ECD) fused to an Fc domain was validated by enzyme-linked immunosorbent assay (ELISA) (Supplementary Fig. 5). Since the nanoHER2 binding epitope on the HER2 ECD was unknown prior to this study, we simulated its docking to HER2 ECD using AlphaFold-multimer[43–45], predicting an interaction surface on Domain III close to the binding site for AffiHER2 (Supplementary Fig. 6A, B). We then performed competition ELISA, which showed that all 6 Fabs had no effect on nanoHER2 binding to HER2 ECD, but nanoHER2's interaction was efficiently competed out by AffiHER2 (Supplementary Fig. 6C, D).

Each binder was dimerized with itself or another binder using SpyMask methodology (Fig. 3A). A subset of DoubleCatcher-assembled bispecific molecules (one bispecific harboring an N-terminal Fab, nanobody, or affibody, and one bispecific harboring a C-terminal Fab, nanobody, or affibody) were analyzed by mass spectrometry (Supplementary Fig. 7), to confirm their correct assembly and purity. We studied the effects of the resulting matrix of anti-HER2 bispecifics and homodimers on metabolic activity in HER2-addicted BT474 cells (a human ductal breast carcinoma cell-line) or SKBR3 cells (a human adenocarcinoma cell-line), quantifying using the resazurin assay[37].

Following incubation of BT474 or SKBR3 cells with anti-HER2 bispecific assemblies (100 nM) for 96 h, the change in metabolic activity was calculated relative to untreated cells (Fig. 4C, D). We discovered that the different assemblies exhibited a range of potent antagonistic and agonistic activities. Binders with statistically significant activities were identified by two-way analysis of variance (ANOVA) with Dunnett's correction for multiple comparison at $\alpha = 0.05$. We found that bispecific constructs displaying one or more arms binding to domain II (39 S or Pert) are consistently anti-proliferative.

Anti-HER2 bispecifics also showed different potencies in SKBR3 cells relative to BT474 cells. Firstly, a higher number of bispecifics had significant proliferative activity in BT474 (20 constructs) than in SKBR3 (10 constructs). Secondly, constructs with significant anti-proliferative activity were more potent in BT474 (average metabolic activity of significant antagonists = $58.4 \pm 20.4\%$) than in SKBR3 (average metabolic activity of significant antagonists = $64.2 \pm 7.0\%$) (mean $\pm$ 1 s.d., $n = 3$). In BT474 cells, the potent antagonist Tras NoLink:Tras ($74.1 \pm 6\%$) showed no activity when assembled in the opposite orientation: Tras:Tras NoLink ($103 \pm 1\%$) (mean $\pm$ 1 s.d., $n = 3$). Importantly, the same orientation-dependence was observed in SKBR3 cells for the significantly active bispecifics Tras:AffiHER2, nanoHER2:AffiHER2, MF3958:Tras NoLink, Tras:39 S, 39 S:Tras NoLink and Tras:MF3958; assembly of these binders in the opposite orientation removes their significant agonistic or antagonistic activity. The top 5 bispecifics with significant anti-proliferative or pro-proliferative activity were further analyzed by titrating on SKBR3 cells (Fig. 4E, F).

Examining the effect of removing the linker between binder and SpyTag003, the average difference between bispecifics with at least one Tras versus at least one Tras NoLink was only 2%. The similarity between the activities of Tras and Tras NoLink supports the usage of linker-free binder-SpyTag fusions in future screens, where it will be beneficial to maximize rigidity in the bispecific molecule and impart control over target geometries. Together, these results highlight the importance of screening epitope-diverse bispecific binders in multiple formats and orientations.

## DoubleCatcher architecture can be customized

As with most bispecific antibodies, the format of a bispecific binder built using SpyMask may be influenced by the identity of the SpyTag003-linked binder and by the order in which the binders are coupled. However, the overall shape, size, and flexibility of the bispecific molecule may also be determined by the DoubleCatcher module itself. To amplify the format space to be screened, matrices of bispecific antibodies could therefore be developed in three dimensions: varying two binder arms against the targets of interest and, in parallel, varying the DoubleCatcher module. The highest-scoring ColabFold-predicted structure for DoubleCatcher is displayed in Fig. 5A (left), with the position and orientation of the two SpyTag binding sites represented by cyan vectors. Modifications to the DoubleCatcher architecture that change the relative spacing and orientation of the cyan vectors will reflect changes in the structural relationship between the two target-binding arms in the final bispecific.

Modifications to the architecture of DoubleCatcher may be achieved by varying the linker connecting the two SpyCatcher003 moieties. The amenability of DoubleCatcher to linker modification was tested by substituting the Gly/Ser linker with the central helix from the *Bacillus stearothermophilus* ribosomal protein L9[46]. The DoubleCatcher variant with the helical linker is termed 'DoubleCatcher H-

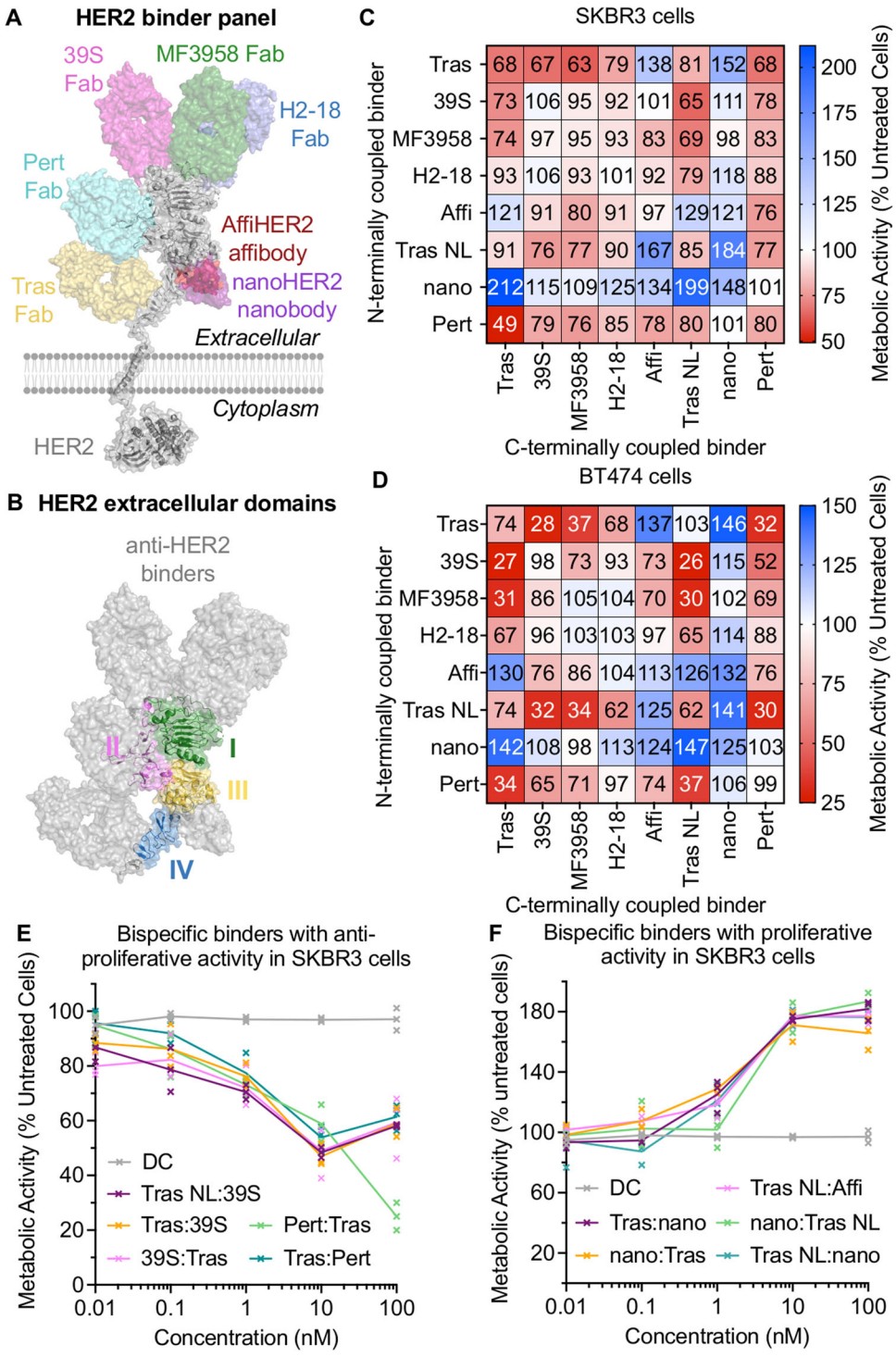

**Fig. 4 | SpyMask bispecific panel exhibits a range of pro- or anti-proliferative activities on cancer cells. A** Binding sites on HER2 for the panel of anti-HER2 binders. For each subdomain (I-IV) of HER2's extracellular domain at least one binding molecule was identified and the cartoon illustrates binding sites on HER2 (gray), with each binder in a different color. Binding sites are derived from crystal structures, except for nanoHER2, which is based on AlphaFold2 prediction and competition ELISA (Supplementary Fig. 6). **B** Subdomains of HER2's extracellular domain are colored, with anti-HER2 binders from (**A**) overlaid in gray. **C** SpyMask anti-HER2 panel effect on SKBR3 cell proliferation. The vertical column indicates the binder at the N-terminal site of DoubleCatcher. The horizontal column shows the binder at the C-terminal site of DoubleCatcher. The effect on proliferation of SKBR3 cells with 100 nM bispecific after 4 days from resazurin assay is shown as a heat map, with red showing reduced proliferation and blue showing enhanced

proliferation, compared to the DoubleCatcher alone control at 100%. Median standard deviation in metabolic activity in anti-HER2 treated SKBR3 cells is 7%. **D** Bispecific panel effect on proliferation of BT474 cells, as in **C**. Median standard deviation in metabolic activity in anti-HER2 treated BT474 cells is 2%. **E** Titration curves for the top 5 bispecifics with significant anti-proliferative activity in SKBR3 cells, as calculated by two-way analysis of variance (ANOVA) with Dunnett's correction for multiple comparison at α = 0.05, assayed as in **C**. **F** Titration curves for the top 5 bispecifics with significant pro-proliferative activity in SKBR3 cells, as calculated by ANOVA with Dunnett's correction for multiple comparison at α = 0.05, assayed as in **C**. For (**E**) and (**F**), individual data points are shown with the line connecting the means (*n* = 3 technical replicates). Source data are provided as a Source Data file.

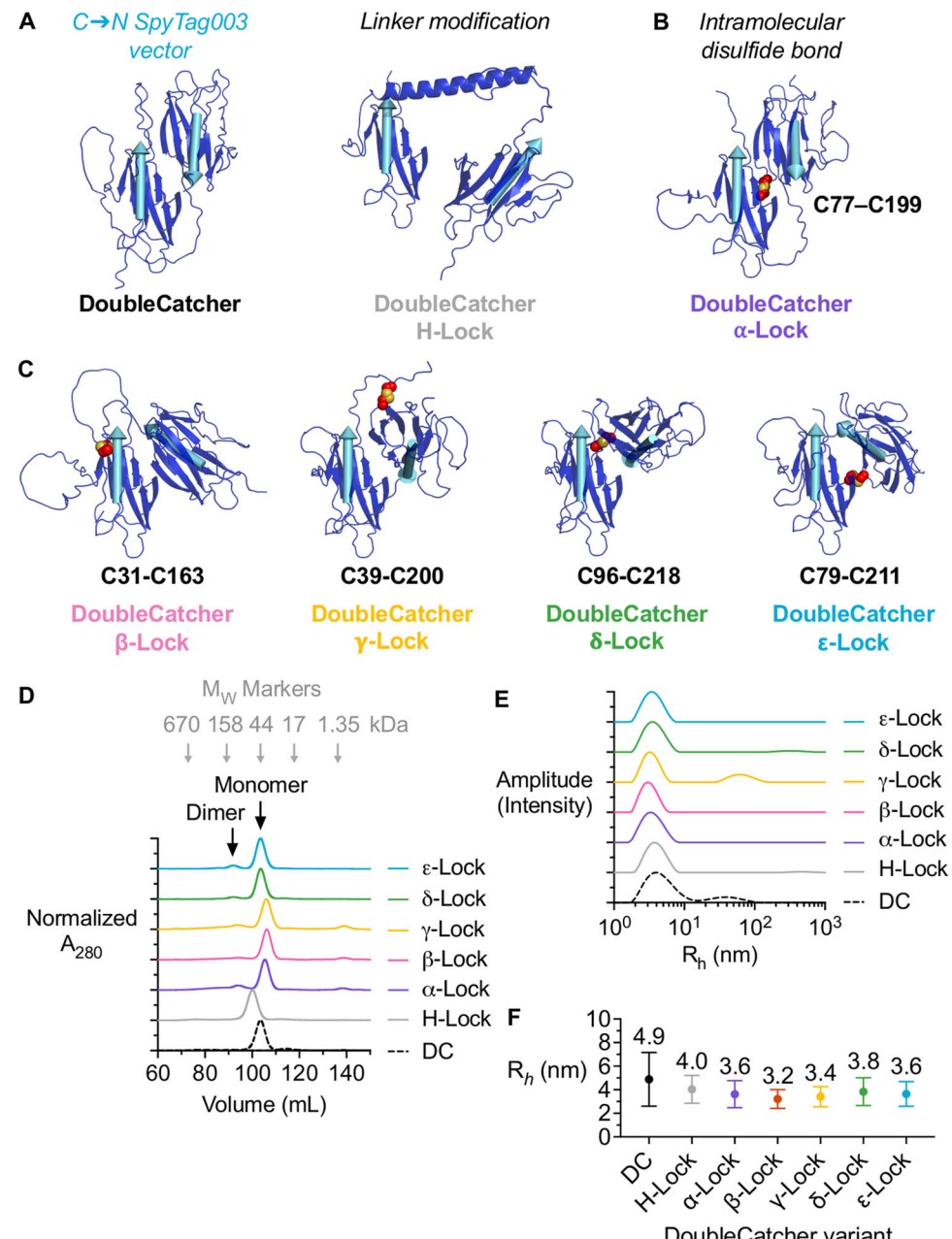

**Fig. 5 | Panel of DoubleCatcher architectures. A** Schematic of DoubleCatcher or DoubleCatcher with helical spacer. AlphaFold2-predicted structures show DoubleCatcher dark blue and the Tag as a cyan vector, indicating the direction that a Fab-SpyTag fusion would project. The flexible linker in DoubleCatcher (left) was replaced with a stable α-helical linker in DoubleCatcher H-lock (right). **B** Schematic of DoubleCatcher locking through an engineered disulfide bond. AlphaFold2-predicted structure, as in **A**. The residues selected for Cys substitution (left) and the disulfide bond in the resulting double-cysteine mutant (DoubleCatcher α-Lock) are labeled as red spheres with sulfur in yellow. **C** Schematic of the other

DoubleCatcher Lock variants, labeled with their Cys substitutions, based on AlphaFold2. **D** Size-exclusion chromatography of DoubleCatcher variants normalized by $A_{280}$. Chromatograms are staggered by 1 from adjacent curves in the y-axis. Apparent $M_w$ was calculated by calibration with $M_w$ standards (gray arrows). **E** Dynamic light scattering of DoubleCatcher variants. Traces are staggered by 1 from adjacent curves in the y-axis. **F** Quantification of dynamic light scattering, showing mean hydrodynamic radius ($R_h$) ± 1 s.d. ($n = 10$ technical replicates) for each DoubleCatcher. Source data are provided as a Source Data file.

lock' (Fig. 5A, right, and sequence in Supplementary Fig. 1). Size-exclusion chromatography analysis of Ni-NTA-purified DoubleCatcher H-lock revealed its apparent molecular mass was 17 kDa larger than DoubleCatcher (Fig. 5D, E), while the predicted increase in mass is only 3.5 kDa, as validated by electrospray-ionization mass spectrometry (Supplementary Fig. 2).

One approach to increasing the rigidity of the DoubleCatcher module that overcomes the flexibility of the SpyCatcher003 termini is to engineer a covalent linkage between the two Catcher domains. One residue from each SpyCatcher003 that resides in a loop region and

close proximity (1.2 nm) in each of the five highest-confidence predicted structures of DoubleCatcher was chosen for cysteine substitution, to engineer a SpyCatcher003-SpyCatcher003 intramolecular disulfide bond. The ColabFold-predicted structure of the double-cysteine variant designed from the top-predicted DoubleCatcher structure is presented in Fig. 5B and is termed 'DoubleCatcher α-lock'. The substantially less variable average distance between Nε of the reactive Lys residues of each SpyCatcher003 in the top 5 ColabFold-predicted structures of DoubleCatcher α-lock (4.7 ± 0.04 nm, mean ± 1 s.d., $n = 10$) suggests the two SpyCatcher003s may have low relative

mobility. The DoubleCatcher double-cysteine variants designed from the four remaining predicted structures of DoubleCatcher are presented in Fig. 5C. Therefore, the combination of SpyTagged antigen binders using these DoubleCatcher variants should provide conformationally distinct bispecific binders that bind targets in different orientations. Size-exclusion chromatography profiles of Ni-NTA-purified double-cysteine DoubleCatcher variants reveal that the construct exists as two species: one main peak corresponding to monomer and one minor peak that corresponds to covalent dimer (Fig. 5D).

DoubleCatcher double-cysteine monomer was incubated with CuSO$_4$ to drive disulfide bond formation prior to downstream analyses. The identity and disulfide bond formation of each DoubleCatcher was validated by electrospray-ionization mass spectrometry (Supplementary Fig. 2). The DoubleCatcher variants were analyzed by dynamic light scattering (DLS) (Fig. 5E), with quantification shown in Fig. 5F. As expected, both the mean hydrodynamic radius (R$_h$) and the mean s.d. of the double-cysteine mutants ($3.5 \pm 1.0$ nm) were less than those for the cysteine-free variants ($4.5 \pm 1.7$ nm) (mean $\pm$ 1 s.d., $n = 10$), consistent with reduced flexibility within the architecture.

### SpyMask-assembled anti-HER2 bispecific binders are sensitive to DoubleCatcher architecture

To investigate the sensitivity of HER2 signaling to dimer geometry, a small-scale matrix of anti-HER2 bispecific binders was assembled by using each of the 7 designed DoubleCatcher variants to dimerize Tras NoLink, nanoHER2 and AffiHER2 in all 9 possible permutations. A subset of these bispecifics was analyzed by mass spectrometry (Supplementary Fig. 8), which confirmed accurate assembly. The assembled bispecific binders were added to HER2-addicted SKBR3 cells (100 nM) for 96 h, after which the change in metabolic activity was calculated (Fig. 6A, B). For each pair of binders, cells were treated with a mixture of the purified binders (100 nM) without DoubleCatcher, to test for monovalent binder activity- 'No DC' matrix in Fig. 6A. Simultaneously, purified DoubleCatcher variants (100 nM) were added to cells, to test for any activity of scaffold alone (Fig. 6A).

In a similar way to the orientation-dependent activity of bispecifics assembled in Fig. 4, we repeatedly found that the potency of specific combinations of binders could be enhanced when assembled onto a different DoubleCatcher variant. The most pronounced example of this is for the pro-proliferative DoubleCatcher β-lock bispecifics containing nanoHER2. Relative to the DoubleCatcher-assembled bispecifics, those bispecifics assembled with DoubleCatcher β-Lock induced 25.6% greater proliferation in SKBR3 cells. Similarly, the anti-proliferative activity of Tras NL:Tras NL ($84.8 \pm 6.8\%$ metabolic activity) was enhanced when dimerized with DoubleCatcher H-Lock ($75.9 \pm 2.5\%$), DoubleCatcher α-Lock ($79.5 \pm 3.9\%$), DoubleCatcher γ-Lock ($80.4 \pm 3.4\%$), DoubleCatcher δ-Lock ($73.6 \pm 4.9\%$), and DoubleCatcher ε-Lock ($72.5 \pm 4.3\%$) (mean $\pm$ 1 s.d., $n = 3$). In addition to increased potency, the ability of some pairs of binders to either agonize or antagonize HER2 signaling was switched when assembled into a different architecture. The bispecific AffiHER2:Tras NoLink showed agonistic activity on the DoubleCatcher scaffold ($112.5 \pm 6.6\%$ metabolic activity), yet the same bispecific assembled using DoubleCatcher H-Lock exhibited antagonistic activity under the same conditions ($82.5 \pm 11.1\%$) (mean $\pm$ 1 s.d., $n = 3$). The switch in activity was even more marked for Tras NoLink:AffiHER2 assembled in DoubleCatcher γ-Lock ($78.6 \pm 3.1\%$) relative to DoubleCatcher ($127.7 \pm 4.1\%$), highlighting the value of screening multiple formats in the assembly of bispecifics (mean $\pm$ 1 s.d., $n = 3$).

### Discussion

Here we have established SpyMask assembly, allowing controlled and precise construction of bispecific binders through sequential SpyTag/SpyCatcher conjugation. The SpyMask platform offers a plug-and-play approach in which the antigen-binding building blocks exist in a common format, fused to the commonly employed SpyTag peptide or its subsequent generations[25,28]. Unlike standard approaches to bispecific assembly[1,2], SpyMask eliminates the need to clone and express each pair of binding proteins into more than one format. Neither SpyTag nor SpyCatcher require any cysteines and the overall assembly is ligated through amide bonds[25], so SpyMask avoids the careful optimization and instability that may arise from bispecific assembly relying on disulfide bond reduction and oxidation. Optimization of SpyMask conditions produced a generalized pipeline that generates bispecifics with up to 95% purity relative to homodimeric by-products. The modularity of the platform invites customization of each constituent, exemplified here through DoubleCatcher variants with different linkers or intramolecular disulfide bonds. There is potential in future work for substantial diversification of the DoubleCatcher module, to generate additional bispecific architectures[47].

In this study, anti-HER2 binders varying in epitope and size were permuted into a matrix of bispecifics. We identified anti-HER2 bispecifics with major differences in their pro- or anti-proliferative activity. Interestingly, this proliferative activity depended not only on the identity of the anti-HER2 monomers, but also on the order of coupling to DoubleCatcher, affirming the importance of screening paratope orientation[37,48]. Furthermore, in assembling a small panel of anti-HER2 binders into bispecifics using DoubleCatchers of varying geometries and flexibilities, we found that the activity or potency of particular pairs of binders is dependent on their scaffold.

Within our panel, we applied two binders related to antibodies in clinical use. The SpyMask platform could be applied together with Fabs based on diverse other clinically approved antibodies, with special interest for those that target signaling such as growth factor receptors (e.g. cetuximab), G protein-coupled receptors (e.g. erenumab) or checkpoint inhibitors (e.g. pembrolizumab). Cellular effects of bispecific bridging depend upon an intricate balance of allosteric interactions between intracellular signaling domains[49], altered heterodimer formation with other receptors[50], and receptor dynamics at the plasma membrane or endolysosomal compartments[37,48]. Therefore, being able to screen large numbers of bispecific combinations in multiple formats is crucial to the discovery of the most potent effector activity. HER2 dimerization is facilitated predominantly by interactions between domain II residues[51], so bispecifics binding there are likely to block HER2 dimerization and activation. Conversely, bispecifics harboring the small, non-domain II binders nanoHER2 (16 kDa) or AffiHER2 (15 kDa) display potent proliferative activity. Indeed, the bispecific constructs with significant agonistic activity in both SKBR3 and BT474 cells all comprised at least one of nanoHER2 or AffiHER2. Where agonistic activity of biparatopic anti-HER2 molecules has been observed, it has been suggested to result from favoring HER2 homo-dimerization over heterodimerization, or from a shorter distance between antigen-binding sites[48,50,52]. Considering their small size, DoubleCatchers coupled to either nanoHER2 or AffiHER2 may bridge HER2 molecules close enough for trans-phosphorylation by HER2 kinase domains. The difference between SKBR3 and BT474 responses may be explained by the differential total expression of HER2, as well as the expression of EGFR and HER3 relative to HER2. BT474 expresses more HER2 ($\sim 2.8 \times 10^6$ copies per cell)[53] relative to SKBR3 ($\sim 1.2 \times 10^6$ copies)[54]. Both express HER3 to a similar level ($1$-$1.2 \times 10^6$ copies) but SKBR3 expresses ~3-fold more EGFR ($2.2 \times 10^5$ copies) than BT474 ($7 \times 10^4$ copies)[55]. HER2-HER3 dimerization mediates activation of PI3K/Akt and the proliferation axis[56], while HER2-EGFR dimers favor MAPK/ERK and pro-survival signaling[57]. Therefore, for bispecifics preventing HER2 homodimerization but not occluding domain II of HER2 (for example, domain I-binding H2-18 or MF3958), heterodimerization may occur more favorably. This could result in a higher frequency of pro-liferative HER2-HER3 dimers in BT474 than SKBR3.

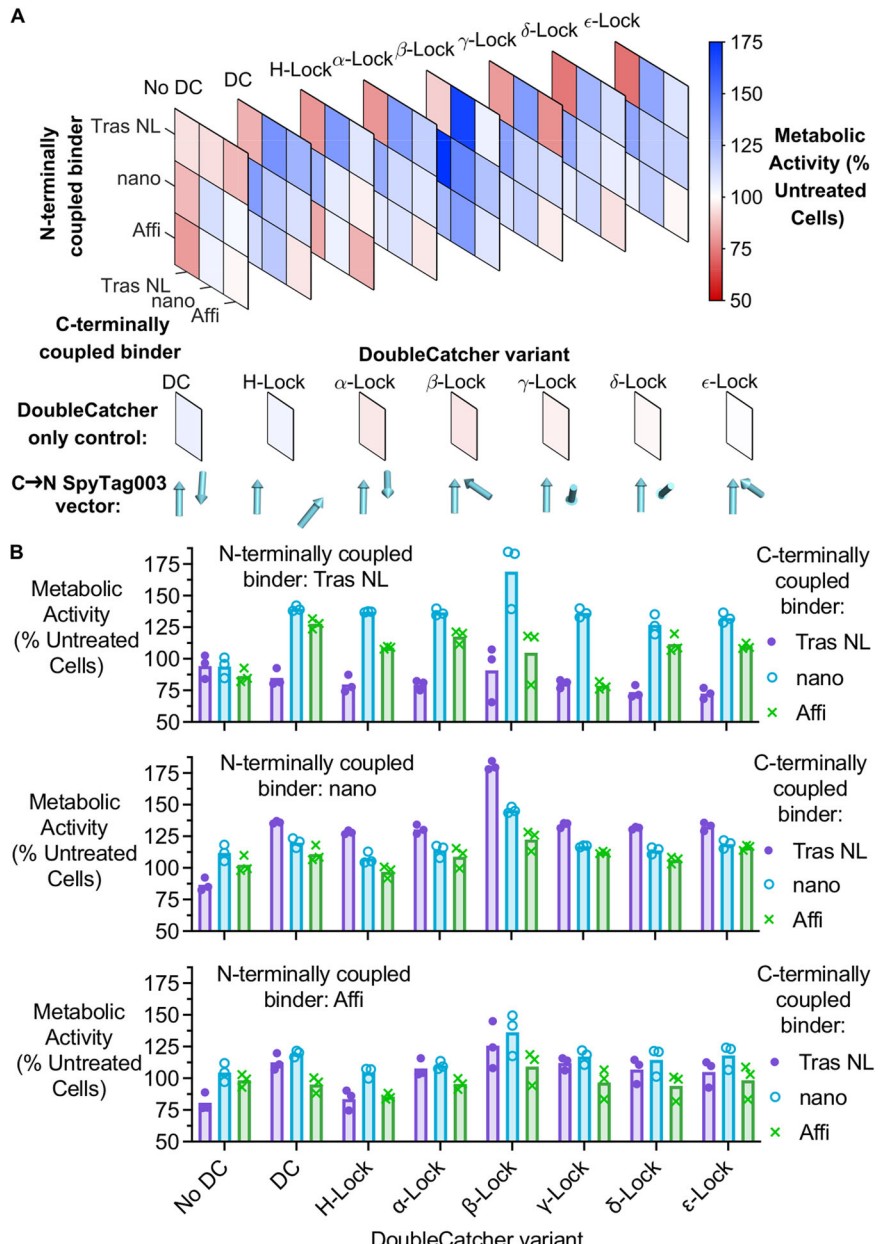

**Fig. 6 | Architectural modifications to anti-HER2 bispecifics change the signaling effects. A** Different DoubleCatcher scaffolds change HER2 signaling. Matrices of anti-HER2 bispecifics were assembled by SpyMask using distinct DoubleCatcher variants. Treatment of SKBR3 cells with the bispecifics (100 nM, 4 days) revealed DoubleCatcher-dependence on their pro- or anti-proliferative potency, as determined by resazurin assay. Mean metabolic activity, relative to untreated control, is presented as stacked heat maps, with red showing reduced proliferation and blue enhanced proliferation. Activity following 100 nM DoubleCatcher variant alone was measured in parallel. Below each variant, vectors depict the position and orientation of the SpyTag003-binding groove from C- to N-terminus within each SpyCatcher003 moiety, such that the binder projects in the direction of the vector arrowhead. **B** Graphical representation of DoubleCatcher-dependence for cell proliferation, following **A**. Each data-point is shown (*n* = 3 technical replicates), with the bar indicating the mean. Source data are provided as a Source Data file.

Differences in activities of anti-HER2 bispecifics have been attributed to the architecture of the molecule, in addition to binder identity. Specifically, this includes the binder's ability to oligomerize HER2 monomers, resulting in arrested receptor diffusion and compartmentalization into membrane nanodomains[37]; the binder-defined distance between cross-linked HER2 molecules[52]; the distortion of HER2 ECD upon binding, such that its ability to homo/hetero-dimerize is affected[50]; and the amenability of the binder's format to increased bispecific-HER2 complex stoichiometries[38]. Our DoubleCatcher panel introduces variation in relative paratope orientation, distance between paratopes, and scaffold rigidity. We hypothesize that these parameters and their combinations each influence HER2 dynamics by one or more

of the mechanisms highlighted. However, further experiments are required to decipher their biochemical behaviors, such as employing Bioluminescence Resonance Energy Transfer (BRET) to study homo- vs. hetero-dimer formation[58], single-particle tracking to quantify receptor diffusion[36], or cryo-electron microscopy to gain insight into conformational homogeneity of different SpyMask-assembled bispecifics[59].

Control over Tag/Catcher activity has previously been achieved using pH[60], redox[61,62], temperature[63], or light[64-66]. While these switches have an array of valuable applications, the simple generation and large change in reactivity of the protease-uncaging system is advantageous for bispecific assembly. In particular, redox changes on the Catcher

would be undesirable for assembling building blocks that themselves contain disulfides, notably nanobodies and various peptide hormones.

Only a small fraction of the surfaceome (estimated at 2,886 different proteins across human cells)[67] has been explored in terms of its potential for bispecific drug generation in academia and industry[2]. Even for an individual member of the surfaceome, different cellular effects are often obtained by binders that recognize different sites[8,52]. Amidst this huge combinatorial space of possible bispecific drugs, the simplicity of SpyMask bispecific assembly may be a valuable tool to identify specific surface proteins to bring together and which particular recognition sites or orientations achieve the most desired cellular response. A potential limitation of SpyMask is that SpyTag/SpyCatcher is derived from *Streptococcus pyogenes*, so immunogenicity may affect the use of DoubleCatcher bispecifics as clinical candidates[68]. Nonetheless, many factors other than being non-self contribute to the immunogenicity of therapeutics[69], e.g. bacterially-derived non-immunoglobulin scaffolds such as affibodies have shown promising results for sustained therapy in clinical trials[70]. For fitting into existing production and purification pipelines[13], the most likely route would be to use SpyMask-derived bispecifics to identify promising hits (including with deep phenotypic analysis by multi-parameter flow cytometry, high-throughput microscopy or RNAseq)[71,72] and then to reformat the binders into a classic Fc bispecific as a lead for future clinical development[2,73].

A major effort is under way to generate recombinant binders to the entire human proteome[22–24]. Therefore, with an arsenal of tens of thousands of binders, methods such as SpyMask to screen the bispecific possibilities will accelerate the understanding and application of synergy for modulating cell signaling. Increasing numbers of functional proteins have been SpyTagged[42,74,75], allowing purification[60,76], immobilization[77], multimerization[78] and networking to other molecular functionalities[25]. Beyond proteins, SpyTag/SpyCatcher has been functionalized with nucleic acids[79], oligosaccharides[80] or small molecules[81], which will allow modular assembly of bispecifics with more diverse activities. We anticipate applications of SpyMask beyond cell signaling, such as enhancing the recruitment of stem cells for tissue repair[82] or optimizing enzyme networks for catalysis[83].

## Methods

### Plasmids and cloning

Constructs were cloned by PCR methods using Q5 High-Fidelity Polymerase (New England Biolabs) and fragments were assembled by Gibson assembly. See Supplementary Data 1 for primer sequences. All constructs were validated by Sanger Sequencing.

pET28a-SpyTag003-MBP (GenBank Accession no. MN433888, Addgene plasmid ID 133450)[28], pET28-SpyTag003-mClover3 (Addgene plasmid ID 133453)[28], pDEST14-SpyCatcher003 S49C (Addgene plasmid ID 133448)[60], pET28-MBP-super TEV protease (Addgene plasmid ID 171782)[26], pET28a-SnoopTag-AffiHER2-SpyTag (N-terminal His$_6$–SnoopTag–anti-HER2 Affibody–SpyTag) (GenBank accession no. KU296975, Addgene plasmid ID 216280)[84], and pET28a-SnoopTag-SpyTag-(AffiHER2)$_3$ (GenBank Accession no. KU296976, Addgene plasmid ID 216281)[33] have been described. pET28a-SUMO-SpyTag003 (N-terminal His$_6$ tag-SUMO protein-SpyTag003) was cloned previously by Dr. Irsyad Khairil Anuar (University of Oxford) (GenBank Accession no. PP341235, Addgene plasmid ID 216282). pDEST14-SpyCatcher003-TEVs-SpyTag003DA ('Masked SpyCatcher0003'; N-terminal His$_6$ tag-SpyCatcher003-TEV protease cleavage site-SpyTag003 D117A, GenBank Accession no. PP341217, Addgene plasmid ID 216283) was derived from pDEST14-SpyCatcher003 (GenBank Accession no. MN433887, Addgene plasmid ID 133447) by incorporating the TEV protease cleavage site (ENLYFQ/G) and SpyTag003 D117A (RGVPHIVMVAAYKRYK)[28] by Gibson assembly. pDEST14-SpyCatcher003-(GSG)$_3$-SpyCatcher003-TEVs-SpyTag003DA [DoubleCatcher; N-terminal His$_6$ tag-TEV protease cleavage site-two SpyCatcher003 moieties connected by a (GSG)$_3$

spacer-TEV protease cleavage site-SpyTag003 DA, GenBank Accession no. PP341218, Addgene plasmid ID 216284] was synthesized as a gene fragment (Integrated DNA Technologies) and inserted into pDEST14. The Gly/Ser spacer in this construct was then replaced with an α-helical spacer (sequence PANLKALEAQKQKEQRQAAEELANAKKLKEQLEK)[46] and this construct was termed DoubleCatcher H-Lock (GenBank Accession no. PP341219, Addgene plasmid ID 216285). Double-cysteine substitution variants of DoubleCatcher were derived from pDEST14-SpyCatcher003-(GSG)$_3$-SpyCatcher003-TEVs-SpyTag003DA by Gibson assembly: DoubleCatcher α-Lock (GenBank Accession no. PP341220, Addgene plasmid ID 216286), DoubleCatcher β-Lock (GenBank Accession no. PP341221, Addgene plasmid ID 216287), DoubleCatcher γ-Lock (GenBank Accession no. PP341222, Addgene plasmid ID 216288), DoubleCatcher δ-Lock (GenBank Accession no. PP341223, Addgene plasmid ID 216289), and DoubleCatcher ε-Lock (GenBank Accession no. PP341224, Addgene plasmid ID 216290). pET28a-MBP$_X$-MBP$_X$-SpyTag003 (His$_6$-MBP$_X$-SSSGGSGGGSG linker-MBP$_X$-SpyTag003; GenBank Accession no. PP341236, Addgene plasmid ID 216291), where MBP$_X$ is a variant of MBP with stronger maltose binding[33], was cloned by Dr. Henry Wood (University of Oxford). pcDNA3.1-Tras heavy chain-GSG-SpyTag003 ('Tras') (GenBank Accession no. ON131076, Addgene plasmid ID 216292), pcDNA3.1-Tras heavy chain-SpyTag003 ('Tras NoLink') (GenBank Accession no. PP341237, Addgene plasmid ID 216293) and pcDNA3.1-Tras light chain (GenBank Accession no. ON131084, Addgene plasmid ID 216294) were cloned previously by Jamie Lam (University of Oxford) from previous constructs in pOPIN, which have been described[60] and contain the Fab0.11 D102W mutation for higher affinity HER2 binding[39]. pET28-SpyCatcher002-MBP (GenBank Accession no. PP341225, Addgene plasmid ID 216295) was constructed by amplifying SpyCatcher002 from pDEST14-SpyCatcher002 (GenBank Accession no. MF974388, Addgene plasmid ID 102827) and cloning into pET28-SpyTag002-MBP (GenBank Accession no. MF974389, Addgene plasmid ID 102831) using Gibson assembly.

Sequences for the heavy and light chains of anti-HER2 Fabs were synthesized (Twist Bioscience) and cloned into pcDNA3.1 plasmid: 39 S[85] (39 S Heavy- GenBank Accession no. PP341226, Addgene plasmid ID 216296; 39S Light- GenBank Accession no. PP341227, Addgene plasmid ID 216297), MF3958[86] (MF3958 Heavy- GenBank Accession no. PP341228, Addgene plasmid ID 216303; MF3958 Light- GenBank Accession no. PP341229, Addgene plasmid ID 216304), H2-18[87] (H2-18 Heavy- GenBank Accession no. PP341230, Addgene plasmid ID 216307; H2-18 Light- GenBank Accession no. PP341231, Addgene plasmid ID 216308) and Pert[88] (Pert Heavy- GenBank Accession no. PP341232, Addgene plasmid ID 216309; Pert Light- GenBank Accession no. PP341233, Addgene plasmid ID 216310). Each Fab heavy chain had the assembly: N-terminal signal peptide-Heavy variable domain-Human IgG1 C$_H$1 domain-EPKSC IgG1 upper hinge region residues-SpyTag003-His$_6$ tag, with the exception of Pert for which the hinge region was omitted. Each Fab light chain had the assembly: N-terminal signal peptide-Light variable domain-Human kappa light chain. pET28a-nanoHER2-SpyTag003 (GenBank Accession no. PP341234, Addgene plasmid ID 216312), based on the 11A4 nanobody against HER2[40], was cloned with a C-terminal His$_6$ tag.

pHL-HER2 extracellular domain (ECD)-human IgG Fc domain was cloned by Dr Jayati Jain (University of Oxford).

### Bacterial protein expression

pDEST14 constructs were transformed into chemically competent *E. coli* C41 (DE3), a gift from Anthony Watts (University of Oxford), except for the pDEST14-SpyCatcher003-(GSG)$_3$-SpyCatcher003-TEVs-SpyTag003DA disulfide-containing variants, which were transformed into chemically competent *E. coli* T7SHuffle (NEB). pET28a constructs were transformed into chemically competent *E. coli* BL21 (DE3) (Agilent Technologies), with the exception of pET28a-SnoopTag-AffiHER2-

SpyTag, which was transformed into chemically competent *E. coli* T7 Express (NEB). Colonies were picked into 10 mL LB containing either 100 μg/mL ampicillin (pDEST14/pETDuet) or 25-50 μg/mL kanamycin (pET28a) and grown at 37 °C at 200 rpm for 4-16 h. Starter cultures were diluted 100-fold into either 1 L LB supplemented with appropriate antibiotic and 0.8% (w/v) glucose, 1 L LB supplemented with antibiotic alone (T7 Express only), or 2×TY supplemented with antibiotic and 0.5% (v/v) glycerol (T7SHuffle only), and grown at 37 °C with shaking at 200 rpm. At $OD_{600}$ 0.5-0.6, expression was induced with 0.42 mM isopropyl β-D-1-thiogalactopyranoside (IPTG; Fluorochem). pDEST14-transformed cultures were then incubated for 4 h at 30 °C and 200 rpm, while all other cultures were incubated for 16 h at 18 °C and 200 rpm post-induction. Cells were harvested by centrifugation at 4000 $g$ for 15 min at 4 °C and pellets were stored at −80 °C until purification.

## Mammalian protein expression

Anti-HER2 Fabs and HER2 ECD-Fc were expressed in Expi293F cells (Thermo Fisher, A14635), which were maintained in Expi293 Expression Medium (Thermo Fisher) supplemented with 50 U/mL penicillin and 50 μg/mL streptomycin and grown in a humidified Multitron Cell incubator (Infors HT) at 37 °C with 8% (v/v) $CO_2$ and shaking at either 125 rpm (125 mL flasks; Corning) or 240 rpm (50 mL mini bioreactor tubes; Corning). After ≥ 3 passages, cells were diluted to $3 × 10^6$ cells/mL in antibiotic-free Expi293 Expression Medium and transfected using an ExpiFectamine 293 Transfection Kit (Thermo Scientific). ExpiFectamine 293 Transfection Enhancers 1 and 2 were added 20 h post-transfection and cultures were incubated for 5 days post-transfection, when cell viability fell below 50%. Cell supernatants were supplemented with cOmplete, Mini, EDTA-free Protease Inhibitor Cocktail (Roche) and clarified by centrifugation at 4000 g at 4 °C, followed by filtration through a 0.22 μm syringe filter (Thermo Fisher). One-tenth supernatant volume of Ni-NTA wash buffer (50 mM Tris-HCl pH 7.8, 300 mM NaCl, 10 mM imidazole) was added to filtered supernatant, which was then pH-adjusted to pH 7.6 using 1 M Tris-HCl pH 8.0. Supernatants were stored at −80 °C until purification. The typical yield for SpyTag003-fused Fabs ranged from 6 to 156 mg per L of culture.

## Ni-NTA purification of His$_6$-tagged proteins

After thawing, bacterial cell pellets were resuspended in 1× Ni-NTA buffer (50 mM Tris-HCl pH 7.8, 300 mM NaCl) supplemented with cOmplete, Mini, EDTA-free Protease Inhibitor Cocktail (Roche) and 1 mM phenylmethylsulfonyl fluoride (PMSF), and lysed by sonication on ice. SpyCatcher003 S49C variants and DoubleCatcher double-cysteine variants were further supplemented with 10 mM 2-mercaptoethanol to break any disulfide bonds to other proteins. Clarified cell lysates were centrifuged at 30,000 $g$ for 30 min at 4 °C and adjusted to pH 7.6 using 1 M Tris-HCl pH 8.0, before purification by Ni-NTA affinity chromatography (Qiagen). Elution of all proteins was performed in the absence of 2-mercaptoethanol. For anti-HER2 binding proteins to be heterodimerized by DoubleCatcher for treatment of cells, the following washes to remove endotoxin were performed directly prior to elution: 50 column volumes (CV) wash buffer containing 0.1% (v/v) Triton X-114 (50 mM Tris-HCl pH 7.8, 300 mM, 10 mM imidazole), followed by 10 CV wash buffer (50 mM Tris-HCl pH 7.8, 300 mM). Clarified mammalian supernatants were purified identically. Eluted proteins were dialyzed three times against PBS pH 7.4 (137 mM NaCl, 2.7 mM KCl, 10 mM $Na_2HPO_4$, 1.8 mM $KH_2PO_4$) in 3.5 kDa molecular weight cut-off Spectra/Por tubing (Spectrum Labs) at 4 °C. SpyCatcher003 S49C variants to be coupled to resin were instead dialyzed into coupling buffer (50 mM Tris-HCl pH 8.5, 5 mM EDTA). Protein concentrations were determined from $A_{280}$ using their predicted extinction coefficient from ExPASy ProtParam. Typical yield for SpyCatcher003-containing variants was 10-20 mg per L of culture. A typical yield for affibody expression was ~3 mg per L of culture and

30 mg per L for nanobody expressions. Expression of SpyTag003-fused, non-antigen binding proteins yielded 5-15 mg protein per L of culture.

Prior to assembly of bispecific molecules with DoubleCatcher for mammalian cell treatment, endotoxin was removed from DoubleCatcher scaffolds by phase separation with Triton X-114[89]. Purified protein samples were transferred to 1.5 mL endotoxin-free microcentrifuge tubes (StarLab, cat. E1415 − 1510), to which 1% (v/v) Triton X-114 was added. Samples were incubated on ice for 5 min, until all Triton X-114 was dissolved. Samples were then incubated at 37 °C for 5 min and centrifuged for 1 min at 16,000 g at 37 °C. The supernatant was pipetted off, and the entire procedure was repeated twice. Endotoxin concentration of any bispecific constituent to be added to cells was determined using the Limulus amebocyte lysate (LAL) Chromogenic Endotoxin Quantitation Kit (Thermo Fisher) according to the manufacturer's instructions, which confirmed the endotoxin level to be below 1 endotoxin unit/mL.

## SpySwitch purification of SpyTag003-tagged proteins

$(MBP_X)_2$-SpyTag003 cell pellets were lysed and clarified as outlined above, except thawed pellets were resuspended in SpySwitch buffer (50 mM Tris-HCl pH 7.5, 300 mM NaCl). SpySwitch purification was performed at 4 °C, as described previously[60]. Briefly, SpySwitch resin (0.75 mL per 1 L bacterial culture) was equilibrated with 2×10 CV SpySwitch buffer, prior to incubation with clarified cell lysate on an end-over-end rotator for 1 h at 4 °C, after which resin was washed with 4 × 10 CV SpySwitch buffer. pH-dependent elution was performed by incubating the resin with 6 × 1.5 CV SpySwitch elution buffer (50 mM acetic acid, 150 mM NaCl, pH 5.0) for 5 min each, which was immediately neutralized by mixing the flow-through with 6 × 0.3 CV of 1 M Tris-HCl pH 8.0. Proteins were subsequently dialyzed into PBS pH 7.4.

## Polyacrylamide gel electrophoresis

SDS-PAGE was performed using 8-16% polyacrylamide gels in an XCell SureLock system (Thermo Scientific). SDS-PAGE was run at 200 V in 25 mM Tris-HCl pH 8.5, 192 mM glycine, 0.1% (w/v) SDS. Gels were stained with Brilliant Blue G-250 and destained with Milli-Q $H_2O$ prior to imaging on a ChemiDoc XRS+ imager. ImageLab 6.1.0 software (Bio-Rad) was used for densitometric quantification of bands. For imaging on an iBright FL1500 imaging system (Thermo Fisher), analysis was performed using iBright Analysis Software Versions 5.0.1 and 5.2.0 (Thermo Fisher).

## Size-Exclusion Chromatography

Ni-NTA- or SpySwitch-purified protein samples (SpyCatcher003-containing variants, SpyTag003-linked non-antigen binding proteins) were injected onto a pre-equilibrated HiLoad 16/600 Superdex 200 pg column (GE Healthcare) or a HiLoad 16/600 Superdex 75 pg column (SUMO-SpyTag003 and affibodies; GE Healthcare). Samples were run on an ÄKTA Pure 25 (GE Healthcare) fast protein liquid chromatography machine at 1 mL/min in PBS pH 7.4 at 4 °C. The absorbance profile of column elutions was recorded at 230 nm, 260 nm, and 280 nm. Each column was calibrated using 660 kDa to 1.35 kDa molecular weight protein standards (thyroglobulin, IgG, ovalbumin, myoglobin, and vitamin B12) (Bio-Rad). Following peak analysis by SDS-PAGE, fractions containing purified protein at the expected molecular weight were concentrated using Vivaspin 20 centrifugal filters (Cytiva) at 4 °C and then stored at −80 °C.

## Resin coupling

SpyCatcher003 S49C was attached to SulfoLink Coupling Resin (Thermo Fisher) according to the manufacturer's instructions, with the following changes: SpyCatcher003 S49C in coupling buffer was concentrated to 20 mg/mL using a Vivaspin 20 5 kDa molecular weight cut-off centrifugal filter at 4 °C, reduced with 1 mM tris(2-carboxyethyl)

phosphine (TCEP) for 30 min at 25 °C, and coupled to SulfoLink Coupling Resin at 20 mg reduced protein per mL of packed resin. Following washes with 1 M NaCl, the coupled resin was washed with TP buffer (25 mM orthophosphoric acid adjusted to pH 7.0 with Tris base) to remove SpyCatcher003 S49C that had bound non-covalently to the resin. The coupled resin was stored in 20% (v/v) ethanol in PBS pH 7.4 at 4 °C for ≤ 3 months.

## Isopeptide bond reconstitution reactions

All reactions were carried out in triplicate in PBS pH 7.4. 2.5 μM Spy-Catcher003 (at 25 °C), Masked SpyCatcher003, or DoubleCatcher variants (each at 37 °C) was reacted with 5 μM SpyTag003-linked protein. At each time-point or end-point indicated, reactions were quenched by adding 6× SDS loading buffer [0.23 M Tris-HCl pH 6.8, 24% (v/v) glycerol, 120 μM bromophenol blue, 0.23 M SDS], with subsequent heating at 95 °C for 5 min in a Bio-Rad C1000 thermal cycler. Reactions were analyzed by SDS-PAGE and the depletion of SpyCatcher003-derived reactant over time was quantified by densitometry. The extent of reaction between binding partners at each time point is defined as $100 \times [1 - (SpyCatcher003\text{-}derived\ reactant)/(average\ density\ of\ SpyCatcher003\text{-}derived\ reactant\ in\ control\ lanes)]$.

## ELISA

To validate binding activity of anti-HER2 binders to the HER2 ECD, 80 nM SpyCatcher002-MBP in PBS pH 7.4 was adsorbed onto clear flat-bottom Nunc Maxisorp 96-Well Plates (Thermo Fisher, cat. 442404) for 16 h at 4 °C. Wells were washed three times with PBS pH 7.4 with 0.1% (v/v) Tween-20 (PBS-T), before blocking in PBS pH 7.4 with 5% (w/v) bovine serum albumin (BSA) (blocking buffer) for 1 h at 25 °C. Wells were washed three times with PBS-T, after which 80 nM of the specified SpyTag-coupled anti-HER2 binder (Tras, 39 S, MF3958, H2-18, nanoHER2, Tras NoLink, affiHER2 or Pert) or SpyTag003-MBP (as a non-specific control) in blocking buffer were added and incubated for 1 h at 25 °C. For the no binder control wells, blocking buffer alone was added. After wells were washed three times with PBS-T, serial dilutions of HER2 ECD-Fc in blocking buffer were incubated with binder-coated wells for 1 h at 25 °C. Wells were washed again three times with PBS-T, before incubating with 1:5,000 mouse anti-human IgG Fc conjugated with HRP (GenScript, cat. A01854) in blocking buffer for 1 h at 25 °C. Six final washes were performed with PBS-T, before 1-Step™ Ultra TMB-ELISA Substrate Solution (Thermo Fisher) was added to the wells for 5 min. Reactions were quenched with 1 M HCl and $A_{450}$ was measured using a FLUOstar Omega plate reader. All ELISA experiments were performed in triplicate.

To elucidate the HER2 epitope recognized by nanoHER2 by competitive ELISAs, ELISA assays were performed as described above, with the following changes: wells coated with SpyCatcher002-MBP were incubated with 80 nM nanoHER2 in blocking buffer for 1 h at 25 °C. During this step, 10 nM HER2 ECD-Fc was incubated with serial dilutions of each other anti-HER2 binder (except nanoHER2) in blocking buffer for 1 h at 37 °C. After wells were washed three times with PBS-T, the HER2 ECD-Fc/binder mixtures were added to wells for 1 h at 25 °C. Wells were washed again three times with PBS-T, before incubating with 1:5,000 mouse anti-human IgG Fc conjugated with HRP and the assays were completed as described above.

## Assembly of dimeric molecules using DoubleCatcher

With the exception of reconstitution assays, the assembly of bispecific molecules from two SpyTag(003)-linked proteins of interest, 'Protein A' and 'Protein B', was performed in 100 μL total volume per well in 96-well cell culture plates (Thermo Scientific, cat. 167008). 1) DoubleCatcher variant with concentration $N$ (here, $N = 1–75$ μM) was first mixed with $1.25 \times N$ Protein A-SpyTag003 for 2 h at 25 °C with shaking at 550 rpm. 2) $5 \times N$ Protein B-SpyTag003 and $1/12 \times N$ MBP-superTEV were added directly to the reaction mixture and incubated for 4 h at 34 °C with shaking at 550 rpm. 3) reaction mixture was incubated with

SulfoLink resin-immobilized SpyCatcher003 S49C for 16 h at 4 °C with shaking at 550 rpm, to remove excess unreacted SpyTag003-linked binders from solution. All coupling reactions were carried out in PBS pH 7.4.

## Dynamic light scattering

Size-exclusion chromatography-purified protein samples were centrifuged for 30 min at 16,900 g at 4 °C to remove any aggregates. Samples were filtered through 0.33 mm diameter sterile syringe filters with a 0.22 μm hydrophilic polyvinylidene difluoride (PVDF) membrane (Sigma-Aldrich) and diluted to a final concentration of 25 μM for protein samples > 35 kDa, or 75 μM for samples < 35 kDa into sterile-filtered PBS pH 7.4. 20 μL diluted sample was loaded into a reusable quartz cuvette and measurements (10 scans of 10 s each) were recorded at 20 °C using an Omnisizer (Viscotek). Data were analyzed in duplicate using OmniSIZE 3.0. For each sample, the radius of hydration ($R_h$) was plotted with ± 1 s.d. error bars.

## Cell culture

SKBR3 cells were from ATCC (HTB-30). BT474 cells were from Cancer Research UK, Lincoln's Inn Fields. SKBR3 cells were grown in complete DMEM: Dulbecco's Modified Eagle Medium–high glucose (DMEM) supplemented with 10% (v/v) fetal bovine serum (FBS), 100 U/mL penicillin and 100 μg/mL streptomycin (1 × pen/strep; Sigma-Aldrich), 1% (v/v) GlutaMAX (Thermo Fisher) at 37 °C and 5% (v/v) CO₂. BT474 cells were grown in complete RPMI: RPMI-1640 supplemented with 10% fetal bovine serum (FBS), 100 U/mL penicillin and 100 μg/mL streptomycin (1 × pen/strep; Sigma-Aldrich), and insulin (5 μg/mL; Sigma-Aldrich) at 37 °C and 5% (v/v) CO₂. Cells were passaged at 70-80% confluency and were sub-cultured for fewer than 3 months. Cell-lines were validated as mycoplasma-negative by PCR.

## Metabolic Activity Assay

BT474 or SKBR3 cells were seeded into 96-well plates at $1.25 \times 10^4$ cells per well and grown in DMEM supplemented with 0.5% (v/v) FBS, 1% (v/v) GlutaMAX and 50 U/mL penicillin and 50 μg/mL streptomycin (0.5 × pen/strep) at 37 °C for 24 h. 100 nM and subsequent serial dilutions of anti-HER2 binders coupled to DoubleCatcher were prepared in 125 μL DMEM with 0.1% (v/v) FBS, 1% (v/v) GlutaMAX and 0.5 × pen/strep. Spent media was removed and protein solutions were added to cells. DoubleCatcher or MBP-superTEV alone were applied to cells at the same concentrations as the anti-HER2 binders as a negative control for cell killing. Cells were treated for 96 h at 37 °C with 5% (v/v) CO₂, after which 40 μL 0.15 mg/mL Resazurin (Alamar Blue; Sigma-Aldrich) was prepared in PBS pH 7.4 and sterile-filtered through a 0.22 μm pore-sized membrane filter, before adding directly to each well. Plates were incubated for 4 h at 37 °C with 5% (v/v) CO₂. Fluorescence ($\lambda_{ex}$ 544 nm, $\lambda_{em}$ 590 nm) was measured using a FLUOstar Omega plate reader (BMG Labtech). The fluorescence intensity of media, without cells, containing Resazurin ('background') was subtracted from each measurement. The percent metabolic activity relative to untreated cells was then defined as $100 \times$ (background-subtracted fluorescence of cells)/(background-subtracted fluorescence of untreated cells). Untreated cells are defined as cells grown in DMEM + 1% (v/v) FBS, 0.5 × pen/strep alone.

## Mass Spectrometry

Purified protein constructs, with the exception of DoubleCatcher γ-Lock, SpyTag-(AffiHER2)₃, and Pert-SpyTag003, were analyzed by intact protein mass spectrometry in positive ion mode using a RapidFire 365 jet-stream electrospray ion source (Agilent) coupled to a 6550 Accurate-Mass Quadrupole Time-of-Flight (Q-TOF) (Agilent) mass spectrometer. Each sample was analyzed once. 50 μL protein samples at 10 μM in PBS pH 7.4 were prepared on a 384-well polypropylene plate (Greiner). Samples were acidified to 1% (v/v) formic acid, before aspiration under vacuum for 0.4 s and loading onto a C4 solid-phase extraction cartridge.

Following washes with 0.1% (v/v) formic acid in water (1.5 mL/min flow rate for 5.5 s), samples were eluted to the Q-TOF detector with deionized water containing 85% (v/v) acetonitrile and 0.1% (v/v) formic acid (1.25 mL/min flow rate for 5.5 s). Data were analyzed using Mass Hunter Qualitative Analysis software B.07.00 (Agilent); protein ionization data were deconvoluted using the maximum entropy algorithm. Expected molecular weights for full-length proteins were calculated using the ExPASy ProtParam tool, with the N-terminal fMet (bacterial expression) or signal peptide sequence (mammalian expression) removed. Signal peptide cleavage was predicted by SignalP 6.0[90]. −2 Da was calculated for each predicted disulfide bond. The small amount of +178 Da peak relates to gluconoylation, a common spontaneous post-translational modification for proteins overexpressed in *E. coli* BL21[91]. The observed mass of Pert Fab light chain is consistent with S-cysteinylation (+119 Da), as previously observed on human IgG1 kappa light chains[92].

DoubleCatcher γ-Lock, SpyTag-(AffiHER2)₃, Pert-SpyTag003 and all DoubleCatcher-assembled bispecific molecules were analyzed using an Acquity UPLC System coupled to a Waters Corporation Xevo G2-S Q-TOF mass spectrometer. 30 μL protein samples at 1 μM in PBS pH 7.4 were prepared in UPLC autosampler vials (Fisher Scientific). Each sample was analyzed once. The column used for the LC was Acquity UPLC Protein BEHC4 with dimensions 2.1 mm × 50 mm. Samples were run with a flow-rate of 0.2 mL/min and eluted to the Q-TOF detector in deionized water with 0.1% (v/v) formic acid and 5% (v/v) acetonitrile. Data were processed using MassLynx software V4.2 SCN 971 (Waters) for Waters mass spectrometry systems; protein ionization data were processed using Waters' maximum entropy deconvolution algorithm (MaxEnt 1). For DoubleCatcher-assembled bispecific molecules, the loss of two water molecules (−36 Da) was factored into the calculation to account for formation of two isopeptide bonds.

## Mass Photometry
Microscope coverslips (24 × 50 mm, Menzel Gläser) were cleaned by sequential sonication in 50% (v/v) isopropanol and Milli-Q $H_2O$ (5 min each), dried under a clean nitrogen stream, and assembled into flow chambers with isopropanol/Milli-Q $H_2O$-rinsed and nitrogen-dried silicone gaskets (6 mm × 1 mm, GBL103280, Grace Bio-Labs). Data were acquired on a Refeyn TwoMP mass photometer. Immediately prior to mass photometry, protein samples were diluted into freshly prepared and degassed MP sample buffer [PBS pH 7.4, 300 mM NaCl, 0.5% (v/v) glycerol, 1 mM $NaN_3$, sterile-filtered through a 0.22 μm pore-size syringe filter (Thermo Fisher)]. A dynamin protein mass standard (MS1000 20-40×), diluted into MP sample buffer, was used to calibrate the mass photometer each time a new cover slip was used[93]. The focal position of the microscope was found in flow chambers containing 20 μL sample, after which 0.5-10 μL of MP sample buffer was replaced with an equivalent volume of protein sample, to a final working concentration of 2-50 nM, to account for differences in the dissociation characteristics of different proteins. After ≤ 10 s, images were acquired for 60 s at 331 Hz. Each sample was measured at least twice independently. All data were processed using DiscoverMP v1.2.3 software (Refeyn)[93]. Masses were plotted as mass histogram (bin width = 2.5 kDa) and fitted to Gaussian non-linear regression curves to identify the mean masses of each protein subpopulation within the heterogeneous sample in GraphPad Prism 9 (GraphPad Software). Expected $M_w$ values for protein constituents were calculated using ExPASy ProtParam, with the N-terminal fMet (bacterial expression) or signal peptide sequence (mammalian expression) removed.

## Data Analysis and Graphics Visualization
Data visualization and statistical tests were performed using GraphPad Prism 9 (GraphPad Software) and MATLAB R2023a (Mathworks). Protein structures were visualized using PyMOL 2.5.4 (Schrödinger). The structure of HER2 was assembled in PyMOL from PDB IDs 1N8Z (extracellular domain)[94], 2KS1 (transmembrane domain)[95], and 3PP0

(kinase domain)[96] by connecting the termini of adjacent domains as previously[97]. Binder structures are based on the following: Tras by PDB ID 1N8Z[94]; Pert by PDB ID 1S78[98]; 39 S by PDB ID 6ATT[85]; MF3958 by PDB ID 5O4G[86]; H2-18 by PDB ID 3WLW[87]; and AffiHER2 by PDB ID 3MZW[99]. Protein structures for DoubleCatcher variants were predicted using ColabFold versions 1.3 and 1.5.2[100], and validated using AlphaFold version 2.3.1[43,45]. Structures for HER2 ECD in complex with nanoHER2 were predicted using AlphaFold-multimer[43–45]. Structural predictions were performed using the full-length protein sequence. For all ColabFold-simulated structures of DoubleCatcher, the structure predicted with the highest confidence is presented. For AlphaFold-multimer predicted structures, models with ≥ 2 nanobody loops predicted to contribute to the nanoHER2-HER2 ECD interaction interface were selected and overlaid.

To simulate vectors that reflect the directionality of SpyTag003, the PyMOL modevectors tool was used to connect Ser57 to His53 and Ser179 to His175 by their carbonyl main-chain oxygens[101] (DoubleCatcher numbering, according to sequence listed in Supplementary Fig. 1). The arrowhead represents the N-terminus of the SpyTag003 peptide, such that a binder with a C-terminal SpyTag003 would project from the arrowhead of the vector when coupled to DoubleCatcher.

## Statistics and reproducibility
Statistical significance for metabolic activity assays were calculated by two-way analysis of variance (ANOVA) with Dunnett's correction for multiple comparison at α = 0.05. For representative SDS-PAGE (Figs. 1C, E and 2B, E; Supplementary Fig. 4A, B), observations were confirmed at least once with similar or identical conditions. For Fig. 3D, E, heterodimer assembly and subsequent analyses by mass photometry and SDS-PAGE were repeated at least twice with similar results. For Figs. 4C, D and 6A, B the bispecific assemblies and subsequent metabolic activity assays were repeated at least twice independently with similar results. For ELISA experiments (Supplementary Figs. 5, 6D), triplicate assays were confirmed at least twice with similar results. For mass spectrometry analyses (Supplementary Figs. 2, 3, and 8), experiments were performed once, and for Supplementary Fig. 7 experiments were repeated once with independently-assembled DoubleCatcher heterodimers with identical results. No statistical method was used to predetermine sample size. No data were excluded from the analyses. The experiments were not randomized. The investigators were not blinded to allocation during experiments and outcome assessment.

## Reporting summary
Further information on research design is available in the Nature Portfolio Reporting Summary linked to this article.

## Data availability
Amino acid sequences of Masked SpyCatcher003-containing variants are available in Supplementary Fig. 1. Sequences of other constructs have been deposited in GenBank as described in the section Plasmids and Cloning. Plasmids encoding DoubleCatchers and related binders have been deposited in the Addgene repository (https://www. addgene.org/Mark_Howarth/) as described in the section Plasmids and Cloning. All raw mass spectrometry data (from 33 runs) have been deposited to the ProteomeXchange Consortium via the PRIDE[102] partner repository with the dataset identifier PXD049393. Further information and request for resources and reagents should be directed to and will be fulfilled by the lead contact, M.R.H.. Source data are provided with this paper.

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

## Acknowledgements

A.H.K. and M.R.H. were funded by the Biotechnology and Biological Sciences Research Council (BBSRC BB/S007369/1). C.L.D. was supported by the BBSRC Oxford Interdisciplinary Bioscience Doctoral Training Partnership (DTP) (grant number BB/T008784/1). We thank Dr. David Staunton from the University of Oxford Department of Biochemistry Biophysical Suite for help with biophysical analysis. We thank Dr. Anthony Tumber of University of Oxford Department of Chemistry for assistance with RapidFire Mass Spectrometry, supported by the BBSRC (BB/R000344/1). For the purpose of Open Access, the author has applied a CC BY public copyright licence to any Author Accepted Manuscript (AAM) version arising from this submission.

## Author contributions

C.L.D. performed all experiments. A.H.K. designed and purified initial constructs. C.L.D. and M.R.H. designed the project. C.L.D. and M.R.H. wrote the manuscript. All authors read and approved the manuscript.

## Competing interests

M.R.H. is an inventor on a patent on spontaneous amide bond formation (EP2534484) and a SpyBiotech co-founder and shareholder. M.R.H. and A.H.K. are inventors on a patent on SpyTag003/SpyCatcher003 (UK Intellectual Property Office 1706430.4). C.L.D., A.H.K. and M.R.H. are inventors on a patent application pertaining to the bispecific approach described here (United Kingdom Patent Application No. 2313175.8). The authors declare no other competing interests.
