## [Peer Review File · Nature Communications]

REVIEWER COMMENTS

Reviewer #1 (Remarks to the Author):

The authors describe a method to generate sets of bispecific antibodies using a modified (blocking/deblocking) spycatcher/tag technology. They demonstrate that large sets of bispecific antibodies presenting exemplarily as different Her2-targeting binder combinations can be generated. They show that the composition of bispecifics is relevant for functionality. The data are technically sound and the results of the activity assays are conclusive.

The results are, however, neither new nor surprising. The observation that many different binder-combinations and spatial orientations need to be screened to identify best performing entities is state of the art in bispecific antibody development for years.

While the work is technically sound and novel for the spycatcher-aspect, I am a bit puzzled by their described rationale for developing this additional technology. They postulate 'major problems' for 'the most common routes to generate bispecifics' and support this statement by publications which apparently do not represent the current state of the art. Today, generation of large sets of binder combinations by different approaches (direct cloning, Fab or Fc chain exchange, reduction/re-oxidation etc.) are applied in a robust manner (incl. automation) within the industry.

Reviewer #2 (Remarks to the Author):

Reviewing – SpyCombinator Assembly of Bispecific Binders

In this manuscript, the authors describe the generation of bispecific molecules using the DoubleCatcher technology. Here for the generation of bispecific molecules, a masked DoubleCatcher that blocks the binding of the first molecule to one Catcher, but allows the recovery of the molecule to the unmasked Catcher. Using site-specific proteases that provides the unmasking of the masked Catcher, a second molecule is able to bind to the DoubleCatcher. This procedure works quite quickly and generates a high formation of bispecific molecules using either antibody fragments, e.g. Fab and affibodies, and/or other molecules, e.g. maltose-binding protein (MBP) and Ubiquitin-like modifier (SUMO). In addition, the authors described the generation of bispecific molecules by using a panel of different anti-HER2 bispecific molecules like Fab, nanobody or affibody binding to different extracellular domains of HER2 receptor. These bispecific molecules were analyzed on two different tumor cell lines testing the anti- or pro-proliferative effect. Furthermore, the modification of the DoubleCatcher either using a flexible linker or introducing disulfide-bond-forming cysteine residues generates different architectures of the DoubleCatcher with different orientation of the coupled molecules. Again, different bispecific molecules

were generated by using a set of anti-HER2 molecules. In this experiment, a change of pro- and anti-proliferative results was detected using the different architectural modification of the DoubleCatcher.

In general, the manuscript is well written and the experiments were described in a clear manner. The idea of using the SpyCatcher technology to generate bispecific molecules is quite interesting and novel in the field of bispecific molecules. The idea of changing the geometry of the scaffold (DoubleCatcher) increases the interest of the idea. Unfortunately, the SpyCatcher is derived from bacteria increasing the potential of immunogenicity, when applied in in vivo experiments. In the manuscript, there are some points that should be more precisely analyzed and/or discussed, which will be listed below in major and minor things.

Major things:

1) The analysis of the bispecific molecules using the unmodified DoubleCatcher looks quite promising (Fig. 4). However, the mass spectrometry analysis of Fab pertuzumab shows not correct folded molecules. In this case, I would like to see different ELISA experiments, using as immobilized antigen all four different extracellular domains of HER2 to show the specificity of the bispecific molecules. Here, I would like to see the specificity of the different bispecific molecules, for example that the bispecific molecule containing trastuzumab and pertuzumab bind to domain II and IV, but not to domain I and III of HER2. In addition to that, I would also like to see an ELISA using the complete extracellular domains (Domain 1 to 4) of HER2 to show the binding strength of the different bispecific molecules. In this case, also include the parental antibodies molecules as control.

2) Using different geometries of the scaffold DoubleCatcher is very good and can solve many questions for different bispecific molecules. Could you also please show the mass spectrometry of these molecules fusing Fab trastuzumab, nano-HER2 and affi-HER2 to the N and C-terminus of the different DoubleCatcher (Fig. 6). In this case, I would also like to see an ELISA as described in point 1) using all four different domains of HER2 as immobilized antigens to show the specificity of the different molecules and using the complete extracellular domain of HER2 as immobilized antigen to show the binding strength of the different bispecific molecules.

3) The mass photometry of CoubleCatcher-SpyTag003-MBP-SUMO-SpyTag003 shows additional peaks, which are not described in the result part. Could you either described this more precisely or show the mass spectrometry of this molecule.

Minor things:

1) As described above, the analysis of the DoubleCatcher in in vivo experiments would strongly increase the impact of the manuscript. Is there any possibility to modify or change the system to reduce the potential of immunogenicity in vivo experiments? I think that analyzing these bispecific molecules (using either different antigen-binding sites or using the different geometry of the DoubleCatcher) in tumor-bearing mice would be very interesting.

2) The idea of using the different geometries of DoubleCatcher for analyzing the potential of different antigen-binding sites in various bispecific antibodies is a great idea. Could you please discuss this issue

more precisely in the discussion, explaining for example which therapeutic antibodies, which are already on the market or in clinical investigation, could fit to the different architectures of the DoubleCatcher. That would increase the impact of the manuscript.

Reviewer #3 (Remarks to the Author):

The manuscript has described the novel method for construction of bispecific binders, which would be valuable for general protein engineers, in the fields of basic science, medical science, and industry. Several methods for construction of bispecific binders have been reported, but the present methods would be more applicable to versatile systems than the previous methods, e.g. enzyme, and antibody. The method has been perfectly mentioned, and the results are sound and reliable. The conclusions are completely supported by the data; therefore, the reviewer recommends this for publication in the journal without further revisions.

One minor point might be the expansion of the present method for therapeutics. The authors could have discussed this method for drug development.

RESPONSE TO REVIEWER COMMENTS

Reviewer #1 (Remarks to the Author):

The authors describe a method to generate sets of bispecific antibodies using a modified (blocking/deblocking) spycatcher/tag technology. They demonstrate that large sets of bispecific antibodies presenting exemplarily as different Her2-targeting binder combinations can be generated. They show that the composition of bispecifics is relevant form functionality. The data are technically sound and the results of the activity assays are conclusive.

The results are, however, neither new nor surprising. The observation that many different binder-combinations and spatial orientations need to be screened to identify best performing entities is state of the art in bispecific antibody development for years.

While the work is technically sound and novel for the spycatcher-aspect, I am a bit puzzled by their described rationale for developing this additional technology. They postulate 'major problems' for 'the most common routes to generate bispecifics' and support this statement by publications which apparently do not represent the current state of the art. Today, generation of large sets of binder combinations by different approaches (direct cloning, Fab or Fc chain exchange, reduction/re-oxidation etc.) are applied in a robust manner (incl. automation) within the industry.

We thank the reviewer for the valuable comments on our manuscript. We appreciate the reviewer's statement that 'the data are technically sound and the results of the activity assays are conclusive'.

What the reviewer presents in quotations does not match the arguments in the original manuscript. Here is the relevant paragraph from the original manuscript:

“The most common routes to generate bispecifics depend upon rearrangement of IgG architecture, particularly through knob-into-hole generation of heterodimeric Fc domains (Spiess et al., 2013), or disulfide bond shuffling in the Fabs (Dengl et al., 2020; Labrijn et al., 2013). Alternative bispecific pairing has been achieved by conjugating the two binders using sortase (Andres et al., 2020), transglutaminase (Plagmann et al., 2009), click chemistry (Sziij and Chudasama, 2021) or split inteins (Han et al., 2017). However, these routes require each binder to be fused to the respective conjugation domain, such that each binder must be re-cloned, re-expressed, and re-purified in two formats to explore all possible bispecific combinations. Such a requirement is a major problem as the field moves towards exploring thousands of binders across the proteome (Cao et al., 2022; Colwill et al., 2011; Dübel et al., 2010).”

Hence we never suggested that industry has any problem to manufacture individual bispecifics, which are obviously made on a large scale very successfully. What we stated was the challenge in expression for taking thousands of binders as one arm and then making thousands of binders cloned on a complementary arm and then using knob-into-hole or disulfide bond shuffling to make well-folded bispecifics with such high diversity. We thank the reviewer for pushing us to clarify the message of the paper. We have now modified this introductory section for further clarity and to direct readers more easily to state-of-the-art references:

- We have added a citation to an excellent review from December 2023 (X. Guo et al. Front Imm 2023), so that readers can find the state-of-the-art on different formats and see the bispecifics from different companies currently in clinical trials.
- The challenges and pitfalls in disulfide bond formation for bispecific generation and production are described in a review by Bristol Myers Squibb, which we now cite (T. Ren et al. Biotech & Bioeng 2021).

- We have added new information with a 2023 reference on the use of combinatorial DNA libraries for bispecific generation.
- There is now no mention of problems and we talk about the opportunity for additional screening approaches.

Here is the relevant section in the current introduction:

“The most common routes to generate bispecifics depend upon rearrangement of IgG architecture, particularly through knob-into-hole generation of heterodimeric Fc domains (Spiess et al., 2013), or disulfide bond shuffling for Fab-arm exchange (Dengl et al., 2020; Labrijn et al., 2013), leading to many important clinical successes (Guo et al., 2023). Alternative bispecific pairing has been achieved by conjugating the two binders using sortase (Andres et al., 2020), transglutaminase (Plagmann et al., 2009), click chemistry (Sziij and Chudasama, 2021) or split inteins (Han et al., 2017). However, these routes require each binder to be fused to the respective conjugation domain, such that each binder must be re-cloned, re-expressed, and re-purified in two formats to explore all possible bispecific combinations. Sophisticated combinatorial DNA libraries have been employed to impart automation into bispecific library production pipeline (Segaliny et al., 2023), although these methods are only suited to single-chain bispecific molecules. Furthermore, while it is widely appreciated that subtle differences in bispecific format parameters can exhibit profound and unpredictable changes in activity (Dickopf et al., 2020), few assembly platforms to date have been designed to optimize more parameters than binder identity and the relative orientation of binders within the bispecific molecule (Madsen et al., 2023; Segaliny et al., 2023; Wu et al., 2017). As the field moves towards exploring combinations of thousands of binders across the proteome (with potentially millions of bispecifics arising), there is an opportunity for new bispecific antibody assembly platforms to allow screening of multiple format parameters simultaneously, without dramatically increasing the manufacturing burden (Cao et al., 2022; Colwill et al., 2011; Dübel et al., 2010).”

The SpyMask strategy provides a new platform to achieves that screening goal through each member of the library only needing to be cloned in a single format, as well as allowing modular validation of each building block.

Secondly the DoubleCatcher engineering in this work provides a novel platform where each binder can be connected for diverse spatial control simply by mixing, which is very different to what is used industrially. The vast majority of bispecifics are regular Fc fusions and then various formats with tetherings at other points around Fc. These limitations in spatial control of current bispecifics are apparent in various reviews where bispecific designs focus on achieving a high level of binder flexibility (U. Brinkmann et al. mAbs 2017; X. Guo et al. Front Imm 2023).

Reviewer #2 (Remarks to the Author):

Reviewing – SpyCombinator Assembly of Bispecific Binders

In this manuscript, the authors describe the generation of bispecific molecules using the DoubleCatcher technology. Here for the generation of bispecific molecules, a masked DoubleCatcher that blocks the binding of the first molecule to one Catcher, but allows the recovery of the molecule to the unmasked Catcher. Using site-specific proteases that provides the unmasking of the masked Catcher, a second molecule is able to bind to the DoubleCatcher. This procedure works quite quickly and generates a high formation of bispecific molecules using either antibody fragments, e.g. Fab and affibodies, and/or other molecules, e.g. maltose-binding protein (MBP) and Ubiquitin-like modifier (SUMO). In addition, the authors described the generation of bispecific molecules by using a panel of

different anti-HER2 bispecific molecules like Fab, nanobody or affibody binding to different extracellular domains of HER2 receptor. These bispecific molecules were analyzed on two different tumor cell lines testing the anti- or pro-proliferative effect. Furthermore, the modification of the DoubleCatcher either using a flexible linker or introducing disulfide-bond-forming cysteine residues generates different architectures of the DoubleCatcher with different orientation of the coupled molecules. Again, different bispecific molecules were generated by using a set of anti-HER2 molecules. In this experiment, a change of pro- and anti-proliferative results was detected using the different architectural modification of the DoubleCatcher.

In general, the manuscript is well written and the experiments were described in a clear manner. The idea of using the SpyCatcher technology to generate bispecific molecules is quite interesting and novel in the field of bispecific molecules. The idea of changing the geometry of the scaffold (DoubleCatcher) increases the interest of the idea. Unfortunately, the SpyCatcher is derived from bacteria increasing the potential of immunogenicity, when applied in in vivo experiments. In the manuscript, there are some points that should be more precisely analyzed and/or discussed, which will be listed below in major and minor things. Major things:

1) The analysis of the bispecific molecules using the unmodified DoubleCatcher looks quite promising (Fig. 4). However, the mass spectrometry analysis of Fab pertuzumab shows not correct folded molecules.

We would like to thank the reviewer very much for the thorough review and for highlighting the importance of introducing geometric diversity into bispecific antibody scaffolds.

Thank you for noting the unexpected MS on the Fab Pertuzumab. We re-sequenced and re-purified our Pert-SpyTag003 construct and we observed a mass change of +116 for light chain as previously (Figure S3A). Following a careful literature review, we identified independent reports of the same mass adduct (+116 Da) on human IgG1 kappa light chains (the identity of the Pert Fab light chain) (Lim, A. et al. 2001 Anal. Biochem. 295, 45–56; Gadgil, H.S. et al. 2006 Anal. Biochem. 355, 165–174) from formation of two internal disulfide bonds (as expected) and S-cysteinylation at the C-terminal free cysteine – a post-translational modification that has a proposed role in regulating light chain stability without impacting binding affinity. Therefore, our observed mass represents cysteinylated Pert-SpyTag003 and we have modified the text to explain this feature. We have also validated the functionality of Pert-SpyTag003, seeing high affinity binding of this Fab to HER2 by ELISA (see below).

In this case, I would like to see different ELISA experiments, using as immobilized antigen all four different extracellular domains of HER2 to show the specificity of the bispecific molecules. Here, I would like to see the specificity of the different bispecific molecules, for example that the bispecific molecule containing trastuzumab and pertuzumab bind to domain II and IV, but not to domain I and III of HER2. In addition to that, I would also like to see an ELISA using the complete extracellular domains (Domain 1 to 4) of HER2 to show the binding strength of the different bispecific molecules. In this case, also include the parental antibodies molecules as control.

HER2 domains are interdependent, with fixed interdomain contacts (HS Cho et al. Nature 2003) and do not fold properly when expressed individually, so such a test is not feasible. There is one paper that expressed Domain IV in insect cells, but showed poor purity, no validation of the folded state beyond circular dichroism, and has not been taken forward in any subsequent paper (S. Kanthala et al. Prot Exp Purif 2016).

The HER2 interaction sites for 6 out of 7 binders in the latest manuscript are validated by previous high resolution crystal structures, which supersedes any specificity analysis that we could do. We list these structures in the Methods: “Binder structures are based on the following: Tras by PDB ID 1N8Z (Cho et al., 2003); Pert by PDB ID 1S78 (Franklin et al., 2004); 39S by PDB ID 6ATT (Oganessian et al., 2018); MF3958 by PDB ID 5O4G (Geuijen et al., 2018); H2-18 by PDB ID 3WLW (Hu et al., 2015); and AffiHER2 by PDB ID 3MZW (Eigenbrot et al., 2010).”

It is highly unlikely that assembly into bispecifics changes which is the contact site on HER2 of a specific high affinity Fab. The focus of this manuscript is not to find the characteristics of current Fabs but to generate a rapid efficient way to combine existing binders.

Nonetheless, we have now performed ELISA to validate each binder-SpyTag003 fusion in a new Fig. S5:

Supplementary Figure 5. Validation of binder panel interaction with HER2 by ELISA. The binding activity of each anti-HER2 binder used in this study to recombinant HER2 ECD-Fc was measured by ELISA. The binder was coupled to the well and then HER ECD-Fc was added at the indicated concentration, before detection with anti-Fc HRP. MBP was an irrelevant protein to serve as a negative control. Each triplicate data point is shown, with the line connecting the mean.

Therefore, we have now validated for Tras, 39S, H2-18, MF3958, AffiHER2, Pert and nanoHER2 that high affinity to HER2 is maintained by these Fabs. Interestingly, HFS2 showed negligible binding to HER2 in the Fab format by ELISA.

Fig. R1: ELISA of HER2 ECD-Fc binding by HFS2-SpyTag003 performed as in Fig. S5, overlapping with the negative control of MBP, and compared to a positive control of Pert-SpyTag003.

This binder, without a previous structure bound to HER2, had much less validation than other binders and was previously only used as a scFv (F. Salimi et al. Mol Biol Rep 2018). Given this uncertainty and that the remaining panel still contains binders to each extracellular domain of HER2, we have focused the revised manuscript on the binders that show good affinity in our format. The binding site of nanoHER2 was previously only supported by an AlphaFold2 structure, but we have added new data consistent with this predicted binding site based on competition ELISA (Fig. S6).

When starting with binders that already have high affinity, the important feature for bispecific combinations is not achieving even more high affinity but rather their signaling activity, which is what we have determined in Fig. 4. Any in vitro test on the binding affinity/avidity of bispecifics would depend to a high degree on the density at which the HER2 is coated on a surface and the surface-tethered HER2 ECD's conformational flexibility, i.e. in vitro factors which do not map well to what happens at the plasma membrane and then later potentially following internalization.

2) Using different geometries of the scaffold DoubleCatcher is very good and can solve many questions for different bispecific molecules. Could you also please show the mass spectrometry of these molecules fusing Fab trastuzumab, nano-HER2 and affi-HER2 to the N and C-terminus of the different DoubleCatcher (Fig. 6).

We have now performed mass spectrometry analysis of DoubleCatcher with Tras NL:nanoHER2, Tras NL:affiHER2, nanoHER2:Tras NL, nanoHER2:affiHER2, affiHER2:Tras NL, and affiHER2:nanoHER2 bispecifics, which is a new Figure S7 and shown below. Given the high M_w range for this electrospray ionization, the observed masses of each bispecific agree well with their expected masses.

Supplementary Figure 7. Heterodimerization by DoubleCatcher was validated by mass spectrometry. Heterodimerization of a subset of anti-HER2 binders (Tras NoLink Fab, nanoHER2 nanobody, and affiHER2 affibody) in each possible combination was performed using DoubleCatcher, following the SpyMask protocol, to generate bispecific binders. The identity of the heterodimer species of the expected molecular weight was confirmed by Tandem Quadrupole Mass Spectrometry, presented within a grid. Rows indicate binder at the N-terminus of DoubleCatcher within the bispecific, and columns indicate the binder at the C-terminus.

For the bispecific construct with nanoHER2 at the N-terminus and affiHER2 at the C-terminus, the bispecific binder was assembled using the full panel of DoubleCatcher variants and the resulting purified assemblies were also analyzed by mass spectrometry in a new Figure S8. The observed masses of each bispecific construct also agreed well with their expected masses.

Supplementary Figure 8. Heterodimerization of nanoHER2 and affiHER2 with panel of DoubleCatcher variants is validated by mass spectrometry. NanoHER2:affiHER2 bispecific binders were assembled using the full panel of DoubleCatcher variants according to the SpyMask protocol. Purified bispecific molecules were analysed by tandem-quadrupole mass spectrometry and the presence of heterodimer at the expected molecular weight was confirmed for all species. Observed masses are indicated above the main peaks and expected masses were calculated from ExPASy ProtParam.

In this case, I would also like to see an ELISA as described in point 1) using all four different domains of HER2 as immobilized antigens to show the specificity of the different molecules

and using the complete extracellular domain of HER2 as immobilized antigen to show the binding strength of the different bispecific molecules.

As discussed above, despite extensive efforts of the field, unfortunately it is not possible to express the four different domains of HER2 in a well-folded state.

3) The mass photometry of CoubleCatcher-SpyTag003-MBP-SUMO-SpyTag003 shows additional peaks, which are not described in the result part. Could you either described this more precisely or show the mass spectrometry of this molecule.

The mass photometry in Fig. 3D shows two dominant peaks, which fit very well to our expected masses for monomeric DC:A:B and monomeric TEV protease. While mass photometry is a valuable tool for identifying dominant species with sufficiently spread masses within a sample, it can be difficult to distinguish minor species that are close in molecular weight, or that are low in concentration. Data that lay outside of the labelled peaks could not be fit to Gaussian distributions by the recommended peak-fitting protocol (Wu, D., and Piszczek, G. Eur. Biophys. J. 2021). Therefore, to identify the unresolved species within the sample, we performed further mass photometry analyses on the individual components. We hypothesize that the wide right shoulder of the peak corresponding to TEV protease in Fig. 3D may be a small fraction of unreacted $(MBP_X)_2$ -SpyTag003 ('B'), with an observed molecular weight of 85 kDa (Figure R2A). While it is possible for the low count histograms between 110-130 kDa to be due to peak overlap, these counts may also be due to the presence of DC:B (Figure R2B). The low intensity signal around ~200 kDa may correspond to a small amount of DC:B:B homodimer (Figure R2B).

Figure R2: DoubleCatcher coupling reaction analysis by mass photometry. (A) MBP_X-MBP_X-SpyTag003 alone was analyzed by mass photometry, showing the presence of a proportion of dimer and trimer in solution, likely because of weak self-association of MBP components (this construct has no cysteines). (B) Formation of a DC:B:B homodimer as analyzed by mass photometry. Major peaks show B alone, DC coupled to a single B, and the DC:B:B homodimer. DoubleCatcher is abbreviated to DC and $(MBP_X)_2$ -SpyTag003 to B.

Given the small size of these shoulders/trace peaks and the uncertainty of fitting and assignment of minor components of a mixture by mass photometry, we think it most sensible not to add specific annotations for this hypothesis in the latest Fig. 3.

Minor things:

1) As described above, the analysis of the DoubleCatcher in *in vivo* experiments would

strongly increase the impact of the manuscript. Is there any possibility to modify or change the system to reduce the potential of immunogenicity in vivo experiments? I think that analyzing these bispecific molecules (using either different antigen-binding sites or using the different geometry of the DoubleCatcher) in tumor-bearing mice would be very interesting.

This is an interesting point. Reducing immunogenicity will require large scale modification of surface residues and potential epitopes for a range of different MHC Class II throughout the SpyTag003/SpyCatcher003 system, as well as probably evolution to restore high stability and isopeptide bond-forming reactivity. This modified system will need extensive validation with sera and PBMCs from a diverse human population. That large project will be pursued in a separate paper.

2) The idea of using the different geometries of DoubleCatcher for analyzing the potential of different antigen-binding sites in various bispecific antibodies is a great idea. Could you please discuss this issue more precisely in the discussion, explaining for example which therapeutic antibodies, which are already on the market or in clinical investigation, could fit to the different architectures of the DoubleCatcher. That would increase the impact of the manuscript.

Thank you for the enthusiasm for this variation in geometry. Indeed, the Trastuzumab (Herceptin) and Pertuzumab (brand name Perjeta) Fabs that we tested here relate to therapeutic antibodies which are already on the market for cancer therapy. The SpyMask platform should be broadly applicable to antibodies in clinical use and trial development. We have modified the Discussion to highlight this point: “Within our panel, we applied two binders related to antibodies in clinical use. The SpyMask platform could be applied together with Fabs based on diverse other clinically approved antibodies, with special interest for those that target signaling such as growth factor receptors (e.g. cetuximab), G protein-coupled receptors (e.g. erenumab) or checkpoint inhibitors (e.g. pembrolizumab).”

Reviewer #3 (Remarks to the Author):

The manuscript has described the novel method for construction of bispecific binders, which would be valuable for general protein engineers, in the fields of basic science, medical science, and industry. Several methods for construction of bispecific binders have been reported, but the present methods would be more applicable to versatile systems than the previous methods, e.g. enzyme, and antibody. The method has been perfectly mentioned, and the results are sound and reliable. The conclusions are completely supported by the data; therefore, the reviewer recommends this for publication in the journal without further revisions.

One minor point might be the expansion of the present method for therapeutics. The authors could have discussed this method for drug development.

We appreciate the reviewer’s comments on the value of the SpyMask platform for research in academia and industry. In the revised manuscript, as requested we have now expanded the discussion to describe further the likely application of SpyMask for drug development:

“Only a small fraction of the surfaceome (estimated at 2,886 different proteins across human cells) (Bausch-Fluck et al., 2018) has been explored in terms of its potential for bispecific drug generation in academia and industry (Guo et al., 2023). Even for an individual member of the surfaceome, different cellular effects are often obtained by binders that

recognize different sites (Mohan et al., 2019; Kast et al., 2021). Amidst this huge combinatorial space of possible bispecific drugs, the simplicity of SpyMask bispecific assembly may be a valuable tool to identify specific surface proteins to bring together and which particular recognition sites or orientations achieve the most desired cellular response. A potential limitation of SpyMask is that SpyTag/SpyCatcher is derived from *Streptococcus pyogenes*, so immunogenicity may affect the use of DoubleCatcher bispecifics as clinical candidates (Rahikainen et al., 2021). Nonetheless, many factors other than being non-self contribute to the immunogenicity of therapeutics (Yachnin et al., 2021), e.g. bacterially-derived non-immunoglobulin scaffolds such as affibodies have shown promising results for sustained therapy in clinical trials (Klint et al., 2023). For fitting into existing production and purification pipelines (Ren et al., 2021), the most likely route would be to use SpyMask-derived bispecifics to identify promising hits (including with deep phenotypic analysis by multiparameter flow cytometry, high-throughput microscopy or RNAseq) (Chattopadhyay et al., 2019; Shin et al., 2019) and then to reformat the binders into a classic Fc bispecific as a lead for future clinical development (Hofmann et al., 2020, Guo et al., 2023).”

REVIEWERS' COMMENTS

Reviewer #1 (Remarks to the Author):

the authors have addressed my comments in a comprehensive manner

Reviewer #2 (Remarks to the Author):

Thank you very much for the analysis of ELISA experiments or mass spectrometry using either Fab molecules or heterodimerized molecules using the SpyCombinator. Very clear and very understandable. Well done and congratulation to the publication of your manuscript.